# Heavy footprints of upper-ocean eddies on weakened Arctic sea ice in marginal ice zones

Georgy E. Manucharyan [1✉] & Andrew F. Thompson [2]

Arctic sea ice extent continues to decline at an unprecedented rate that is commonly underestimated by climate projection models. This disagreement may imply biases in the representation of processes that bring heat to the sea ice in these models. Here we reveal interactions between ocean-ice heat fluxes, sea ice cover, and upper-ocean eddies that constitute a positive feedback missing in climate models. Using an eddy-resolving global ocean model, we demonstrate that ocean-ice heat fluxes are predominantly induced by localized and intermittent ocean eddies, filaments, and internal waves that episodically advect warm subsurface waters into the mixed layer where they are in direct contact with sea ice. The energetics of near-surface eddies interacting with sea ice are modulated by frictional dissipation in ice-ocean boundary layers, being dominant under consolidated winter ice but substantially reduced under low-concentrated weak sea ice in marginal ice zones. Our results indicate that Arctic sea ice loss will reduce upper-ocean dissipation, which will produce more energetic eddies and amplified ocean-ice heat exchange. We thus emphasize the need for sea ice-aware parameterizations of eddy-induced ice-ocean heat fluxes in climate models.

[1] University of Washington, School of Oceanography, Seattle, WA 98195, USA. [2] California Institute of Technology, Environmental Science and Engineering, Pasadena, CA 91125, USA. ✉email: gmanuch@uw.edu

Global warming has led to a dramatic decrease in Arctic summer sea ice extent and thickness, resulting in weaker and more mobile sea ice with stronger fracturing and more lead openings[1–4]. The spatial extent and seasonal migration of marginal ice zones (MIZs)—highly fluctuating areas of relatively weak and low-concentrated sea ice next to the open ocean—are also increasing[5]. On monthly to seasonal timescales, sea ice predictions from climate models show a range of accuracies,[6,7] with sensitive dependence on the initialization month[8]. However, the largest sea ice variability and prediction errors occur within MIZs[9]. On interannual and longer timescales, climate projection models have widespread estimates of sea ice extent[3], which may imply biases in processes that transport heat towards the ice. Various feedbacks associated with longwave radiation, surface albedo, lapse rate, water vapor, clouds[10,11] as well as internal climate variability are among the proposed candidate processes that need to be improved in climate models. This study is focused on exploring mesoscale and submesoscale sea ice-ocean interactions that are not typically resolved in climate models but might substantially affect ocean-ice heat fluxes (OHFs).

Sea-ice growth/melt is highly sensitive to ocean heat fluxes[12], which remain uncertain, especially at meso- and smaller scales. Indeed, a network of ice-mass buoy observations has demonstrated that in areas of dramatic Arctic sea ice loss, the relative importance of ocean heat fluxes on sea ice melt increased over the past few decades[13,14]. Lateral stirring that arises from under-ice Arctic Ocean eddies is far from quiescent, with energetic eddies commonly observed by in situ instruments[15–18]. In MIZs, eddies are visually evident in sea ice concentration patterns[19,20], while sharp meltwater fronts and strong vertical isopycnal excursions are observed by ocean gliders[21] and Ice-Tethered Profilers[22]. Ice mass Buoys have measured sporadic $O(100\,\text{W m}^{-2})$ enhancement in heat fluxes associated with the passage of warm-core eddies in MIZs[22]. Sea-ice evolution also depends on the presence of surface and internal gravity waves that can lead to vertical mixing and affect the floe-size distribution[23–26]. Process studies suggest that ocean fronts and eddies can be formed due to winds[27,28] as well as spatially-heterogeneous buoyancy fluxes from sea ice growth/melt, and are responsible for advecting warm ocean waters laterally and vertically towards the sea ice, accelerating its melt[19,29,30].

With rare exceptions[31], climate models do not resolve ice-ocean interactions at scales comparable to the Rossby deformation radius, typically $O(10\,\text{km})$ in the deep Arctic Ocean[32]. The presence of sea ice, however, can affect ocean turbulence in several critical ways. First, sea ice can frictionally dissipate internal gravity waves[33] and macro-scale ocean turbulence[34–36], affecting mixed layer energetics and the strength of eddy-heat transport. Second, since sea ice can only absorb heat from the ocean that is typically stored beneath the mixed layer, vertical motions in the ocean could lead to an asymmetric response: upwelling is associated with enhanced heat transport whereas downwelling leads to negligible heat transport[37,38]. Thus, the presence of energetic internal gravity waves, vertical mixing, or eddy-induced upwelling would tend to enhance ice-ocean heat fluxes. At the same time, the transmission of stress from fine-scale oceanic features (order of 10 km) to the sea ice cover could cause the sporadic plastic failure of the ice and lead to a reduction in the areal sea ice coverage, which is not captured in low-resolution ocean-ice models[39]. Yet, model parameterizations of mesoscale[40], submesoscale[41], and boundary layer[42] turbulence have been developed for ice-free oceans and hence cannot represent the impacts of changing sea-ice concentration on ice-ocean heat fluxes.

Here, we quantify mechanical and thermodynamical ice-ocean interactions following a seasonal cycle in a global ocean model at a high resolution of <1 km in the Arctic Ocean (see methods).

This unique model includes tides and internal gravity waves, resolves mesoscale eddies in deep Arctic basins, and permits the development of submesoscale flows, thus allowing us to quantify critical sea ice-ocean interactions that are not currently accounted for by climate models. This high-resolution ocean model, simulated for slightly longer than a year, is forced with reanalysis of atmospheric conditions for the year 2012, which corresponds to the record-low sea ice extent[43]. Since the model is not a data assimilation product, it represents one plausible scenario of sea ice development under prescribed atmospheric forcing and hence the regional circulation and sea ice patterns may differ from observations. Nonetheless, the model presents a unique dynamical tool for the exploration of ice-ocean interactions and feedbacks that are not resolved in state-of-art climate models.

## Results

**Strong mechanical sea ice-ocean coupling in mIZs.** Satellite observations of sea ice in MIZs reveal that sea ice clusters in eddies and filaments (see refs. [19,20,44,45] and Supplementary Figure 1). Similar processes are also present in the high-resolution model, which shows a dramatic enhancement in the correlation between ice and ocean vorticity in MIZs, as compared to regions of consolidated ice (Fig. 1). We highlight two distinct regimes of sea ice dynamics manifesting during the summer melt season: (i) high-concentrated sea ice exhibits brittle-type dynamics with abundant linear kinematic features as well as weak vorticity and deformation rates (Fig. 1a, b and Supplementary movie 1); and (ii) low-concentrated sea ice has strong vorticity and deformation rates (Fig. 1a, b), with its concentration field exhibiting patterns similar to a tracer stirred by ocean eddies (Fig. 1a and Supplementary movie 1).

Particularly in mIZs, where the sea ice concentrations are relatively weak and ocean eddies are energetic, the momentum budget, (1), dictates that the magnitude of rheological forces becomes comparable to ice-ocean stresses and Coriolis forces:

$$\underbrace{f\mathbf{k}\times(\mathbf{u_i}-\mathbf{u_o})}_{\text{Coriolis + SSH gradient}} = \underbrace{-(\rho_o/\rho_i)C_d h^{-1}(\mathbf{u_i}-\mathbf{u_o})|\mathbf{u_i}-\mathbf{u_o}|}_{\text{Ice–ocean stress}} + \underbrace{(ch)^{-1}\nabla\cdot\boldsymbol{\sigma}}_{\text{Sea ice rheology}}$$

$$\Rightarrow \underbrace{\frac{\zeta_i-\zeta_o}{f}}_{\text{Weak rheology limit}} \sim \frac{h}{L\,C_d},$$

(1)

where $f$ is the Coriolis parameter and $\mathbf{k}$ is the unit vector in the z-direction, $\mathbf{u_i}$, $\mathbf{u_o}$ are the sea ice and surface ocean velocities, respectively, $\zeta_i$, $\zeta_o$ are their vorticities that scale as $u/L$ where $L$ is the characteristic eddy scale, $\rho_0$ and $\rho_i$ are the reference ocean and ice densities, $C_d \approx 10^{-3}-10^{-2}$ is the drag coefficient[46], $c$ and $h$ are the sea ice concentration and thickness, and $\boldsymbol{\sigma}$ is the internal ice stress tensor[47]. This equation was derived by taking a difference between the momentum equations for the sea ice and ocean mixed layer and omitting the wind stress (see Supplementary Section 3). Such a unique three-way force balance implies a scaling law, Equation (1), that requires the sea ice $\zeta_i$ and mixed layer vorticity $\zeta_o$ to be of comparable magnitude. The non-dimensional parameter $hL^{-1}C_d^{-1} \sim O(0.01-0.1)$ is relatively small given that $\zeta_o f^{-1} \sim O(0.1)$ for mesoscale eddies observed under the sea ice[48]. When the winds are negligibly low, the ice and mixed layer ocean velocities will also be closely linked, following the scaling. However, since winds have a much larger spatial extent than ocean eddies, the atmospheric stress curl provides only a minor contribution compared to the curl of the ice-ocean stresses. Hence the vorticity of free-drifting sea ice in MIZs is expected to be correlated with ocean vorticity, even in the presence of winds. In MIZs, the scaling relationship in (1)

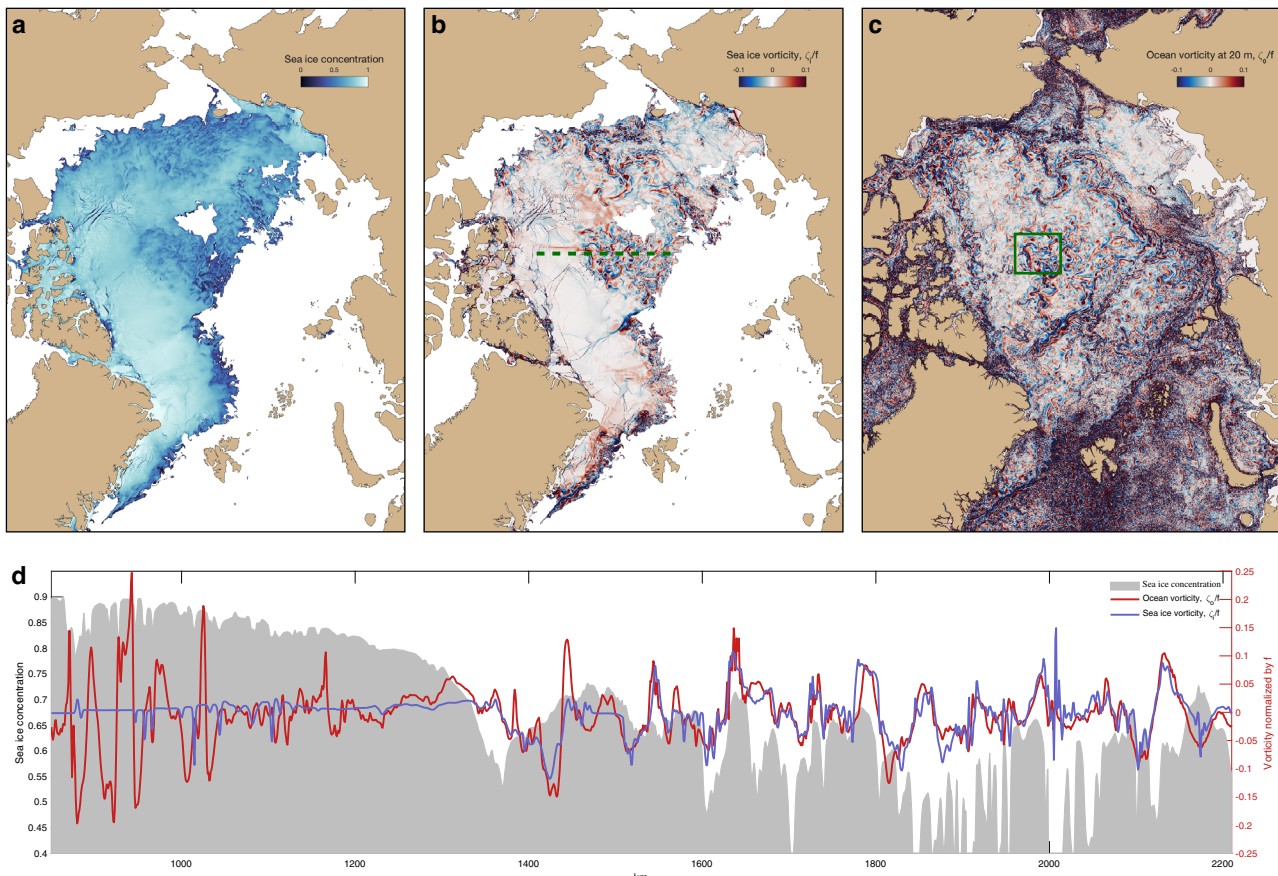

**Fig. 1 State of the sea ice in the Arctic on July 20th, 2012, as simulated by the high-resolution global ocean model. a** Distribution of the effective sea ice concentration, **b** sea ice vorticity $\zeta_i$ normalized by the Coriolis parameter $f$, and **c** ocean vorticity $\zeta_o$ at 20 m depth normalized by $f$; all panels show non-dimensional quantities. White regions in **a**, **b** correspond to the ice-free ocean. The green square in **c** denotes a 400 by 400 km sub-domain used in the subsequent Fig. 3. **d** Sea-ice and ocean properties along the transect across a MIZ (green dashed line in **b**); gray shading denotes sea ice concentration (left $y$ axis), sea ice and upper-ocean vorticity are shown by blue and red curves, respectively (right $y$ axis). Note the strong correlation between the sea ice and ocean vorticity emerging at low sea ice concentrations.

holds, and sea ice exhibits tracer-like behavior, or sea ice is advected passively by the underlying ocean circulation, provided that ice floes are smaller than the characteristic size of eddies.

The free-drift limit of sea-ice dynamics requires sea ice concentration $c$ to be sufficiently small, and we can define a critical concentration $c_{cr}$ that marks the transition to the regime where the scaling in (1) is valid. The magnitude of $c_{cr}$ depends on the details of sea ice rheology parameterization. Specifically for the Viscous-Plastic (VP) rheology used in our numerical simulations, the sea ice strength $P$ is an exponential function of its concentration $P \sim c \exp(-C^*(1-c))$, with an empirically determined parameter $C^* = 20$ strongly affecting the value of the critical concentration $c_{cr}$ (see methods and Supplementary Section 4). The dynamical transition occurs mainly in MIZs at critical concentrations below about $c_{cr} = 0.8$ where the sea ice and ocean vorticity correlation is ~0.7 (Fig. 2a), and the probability distributions of these fields are nearly identical (Fig. 2b). The correlation between sea ice and ocean vorticity is a critical factor that impacts the energetics of upper-ocean eddies via Ekman spin-down (see Equation (2) in methods). By upper-ocean eddies, we imply not only mixed layer eddies and surface-amplified eddies but any eddies with significant surface velocity expressions. Note that the PDFs in Fig. 2 are plotted for a single representative summer day and the critical concentration stays the same regardless of the season. However, the observed fractional area occupied by MIZs (relative to the total sea ice area) is significantly

smaller in winter (~20%) as compared to summer (~70%), with the model substantially underestimating the winter areas but simulating roughly the same fractional MIZ area in the summer (Supplementary Figure 5). The existence of the transitional sea ice concentration and the associated regimes changes are expected to be more important in summer MIZs, which is the time when the model's simulated fractional area agrees with observations.

**Enhancement of OHFs by eddies.** Due to relatively weak vertical mixing, the heat in the Arctic Ocean is efficiently trapped below the mixed layer with near-freezing waters located over warmer halocline water masses. However, localized isopycnal depth perturbations induced by eddies can intermittently bring the subsurface heat into the mixed layer and in contact with sea ice. Anticyclonic halocline eddies have a strong upward displacement of isopycnals in the upper ocean resulting in enhanced ice-ocean heat fluxes, while the opposite is true for cyclonic eddies (Fig. 2d). For example, a typical transect through the MIZ reveals an anticyclone centered at 275 km in Fig. 2d accompanied by enhanced ice-ocean heat fluxes of over 30 W m$^{-2}$, almost an order of magnitude higher than the domain average. While these fluxes might seem large, observations of OHFs reaching O(100 W m$^{-2}$) over an eddy in the Arctic MIZ have been previously reported[22]. The conditional distribution of OHFs is thus positively biased over anticyclones and has a much wider tail corresponding to anomalously strong heat fluxes (Fig. 2c).

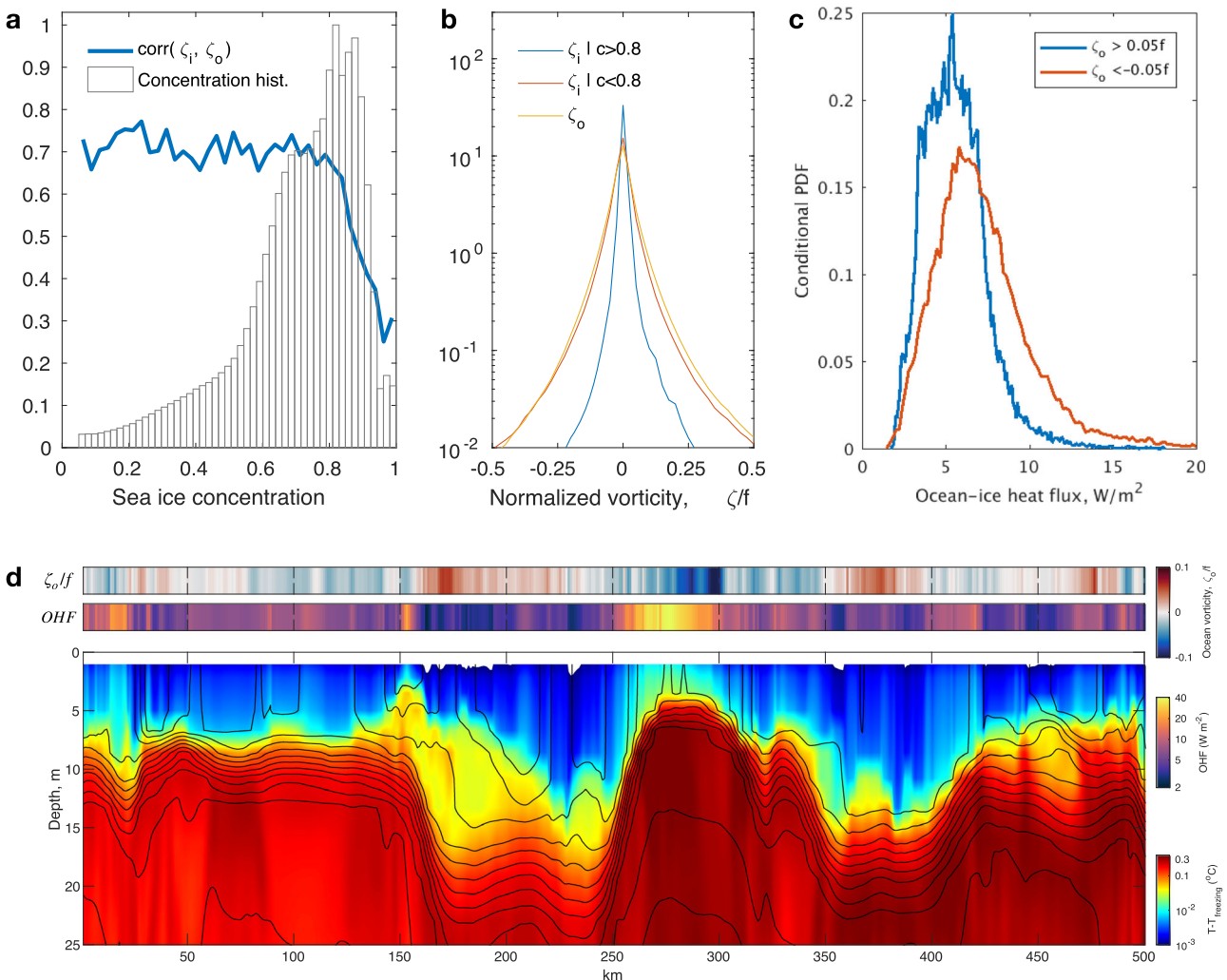

**Fig. 2 The influence of mechanical and thermodynamic ocean processes on sea ice during characteristic summer Arctic conditions, with quantities calculated based on the model state shown in Fig. 1. a** Normalized histogram of sea ice concentrations (shown in bars) and the correlation coefficient between the sea ice and upper-ocean vorticity, $\zeta_i$ and $\zeta_o$, as a function of sea ice concentration (blue curve). An abrupt transition to enhanced vorticity correlations occurs for concentrations below ~80%. **b** Histograms of ocean vorticity (yellow) and sea ice vorticity for concentrations above and below 80% (blue and red curves, respectively). **c** Probability density function of ocean-ice heat fluxes in anticyclones (red) and cyclones (blue). All plotted distributions are derived from the summertime model snapshot shown in Fig. 1 and while the critical concentration stays the same, the histograms of sea ice concentration and the distribution of ocean-ice heat fluxes change substantially depending on the season. **d** The deviation of ocean temperature from freezing along an example transect (taken within the square box region outlined in Fig. 1c) that includes a strong anticyclonic eddy located between 250 km and 350 km. The two horizontal sub-panels show the upper-ocean vorticity (at 20 m depth) normalized by the Coriolis parameter $f$ (top) and the ocean-ice heat fluxes (OHF, middle), along the same transect.

The spatial structure of OHFs in the eddy-resolving ocean model is characterized by many small-scale eddies and filaments (Figs. 2d and 3), representative of frontogenetic and eddying upper-ocean variability. Indeed, considering that the OHF is linearly proportional to the surface ocean temperature above freezing, the heat flux is expected to exhibit filamentary and patchy patterns similar to a surface tracer stirred by mesoscale and submesoscale ocean turbulence. The magnitude of OHFs is typically an order of magnitude higher over strong eddies and filaments, resulting in a fat-tailed distribution of the heat fluxes, especially over anticyclones (Fig. 2c). However, climate models do not account for a substantial part of the ice-ocean heat fluxes induced by eddies and filaments. To compensate for resulting errors in sea ice thickness and extent, other uncertain processes, including both radiative and non-radiative feedbacks, are adjusted[49,50]. As a result, sea ice projections by low-resolution climate models may be biased as they do not represent

potentially key changes in upper-ocean eddy energetics as the climate transitions towards a regime where weakened sea ice occupies a larger area. Below we demonstrate that sea ice can indeed modulate the intensity of mesoscale and submesoscale ocean eddies and corresponding heat fluxes, constituting positive feedback on seasonal timescales.

**Seasonality of heat fluxes, upper-ocean eddies, and eddy dissipation by sea ice.** Due to eddy stirring of surface ocean temperature and salinity, the OHFs manifest as strongly-localized, filamentary structures, being particularly high near anticyclones that are characterized by shoaled isopycnals (Fig. 3). Note that anticyclones are only statistically associated with higher heat fluxes (Fig. 2c) and there are additional diabatic processes that modify the ice-ocean heat fluxes, hence the correlation is not perfect. Nonetheless, the summertime mesoscale eddies are strong and

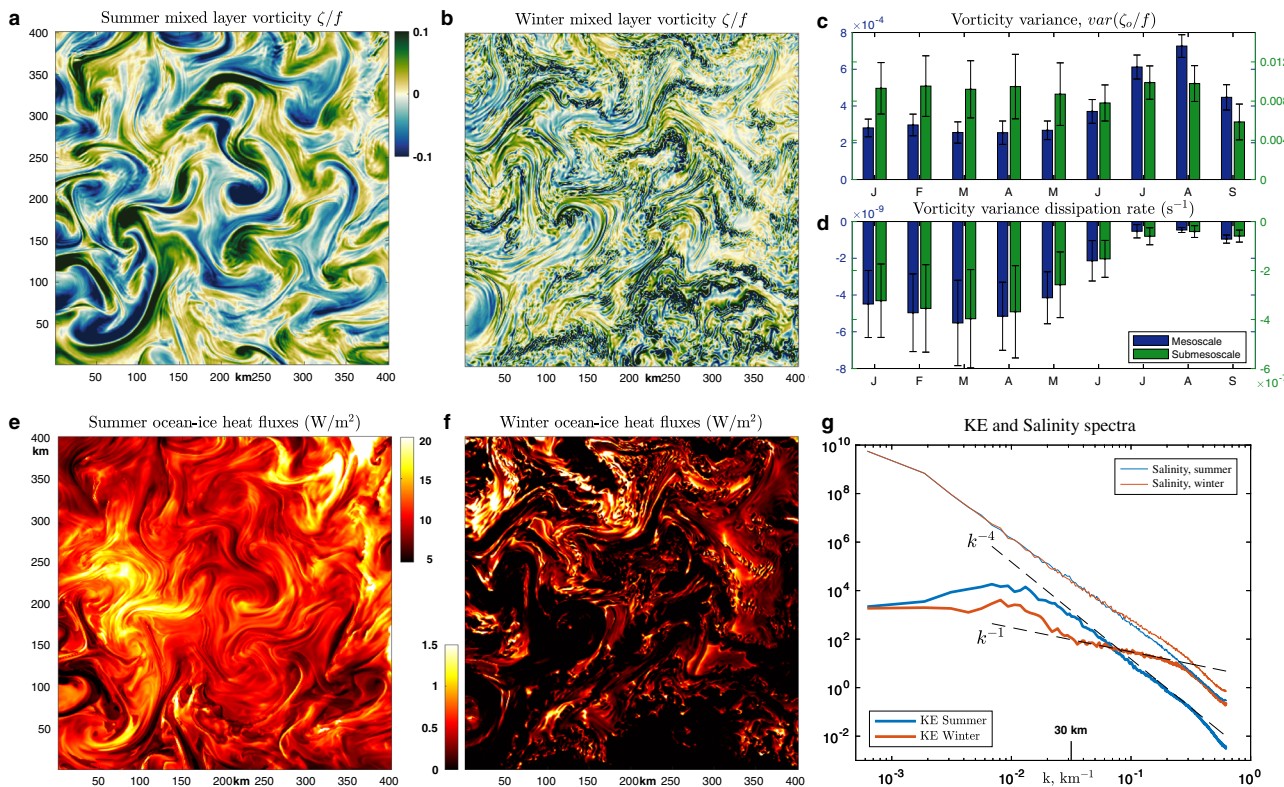

**Fig. 3 Seasonal variations in upper-ocean variability and ocean-ice heat fluxes demonstrating the dominance of small-scale processes in ice-ocean interactions. a, b** Snapshots of ocean vorticity, normalized by the Coriolis parameter $f$, at the end of summer (1 September 2012) and the end of winter (1 March 2012). Maximum values reach ~0.3, but the color bar is saturated at 0.1 to highlight mesoscale features. **e, f** Snapshots of summer and winter ice-ocean heat flux, respectively. The sea ice cover is ~100% for the winter snapshot and ~60% for the summer. **c** Seasonal evolution of the monthly-mean variance of ocean vorticity at 20 m depth normalized by $f$ and **d** monthly-mean and domain-averaged dissipation rate of the ocean vorticity variance (with vorticity also normalized by $f$) due to ice-ocean stress (see Eq. (2)). These values are plotted for flows in mesoscale (blue bars) and submesoscale (green bars) ranges, which are separated by 30-km spatial filtering. The error bars represent standard deviations based on daily variability within a given month. **g** Spectra of near-surface kinetic energy (thick curves) and salinity (thin curves) were diagnosed at 20 m depth for summer (blue colors) and winter (red colors) months. Under-ice-ocean density is governed by salinity and hence its spectrum is closely related to the potential energy spectrum. Dashed curves denote $k^{-1}$ and $k^{-4}$ power-laws for reference. A 30-km wavelength, indicated in **g**, was used to filter mesoscale from submesoscale motions in **c** and **d**.

even in a single snapshot it is not difficult to find many instances of coherent regions of negative mesoscale vorticity being co-located with enhanced OHFs; for example, see elevated heat fluxes near anticyclonic patches at the coordinates (70,40), (230,200), and (350,350) in Fig. 3a, e. In winter, the surface expressions of mesoscale eddies are weaker and the correlation between anticyclonic patches and elevated heat fluxes is less evident, with the heat flux patterns having much smaller length scales associated with submesoscale eddies (Fig. 3b,f). Importantly, in all seasons, the ice-ocean heat fluxes are intermittent and spatially localized, such that a small surface area corresponding to the strongest heat fluxes provides the dominant contribution to the domain-averaged heating. Thus, the magnitude of ice-ocean fluxes not only depends on the availability of sub-mixed-layer heat but also on the energetics and characteristics of upper-ocean meso- and submesoscale eddies. Here we use the term submesoscale to refer to motions that are at scales smaller than the first baroclinic deformation radius. This definition is motivated by spectral representations of the flow, a well-studied diagnostic of ocean turbulence characteristics[51], that show an abrupt transition of the kinetic energy (KE) spectral slope from $k^{-4}$ in the summer to $k^{-1}$ in the winter at ~30-km wavelength (Fig. 3g), where $k$ is the isotropic wavenumber. The very shallow $k^{-1}$ KE spectral slope in winter is consistent with the presence of energetic submesoscale features that are superimposed over larger-scale mesoscale eddies in winter (compare a and b in Fig. 3).

The intensity and length scales of under-ice-ocean eddies and the corresponding ice-ocean heat fluxes are strongly seasonal (Fig. 3). The summer is characterized by a mesoscale eddy field with length scales greater than $O(30 \text{ km})$ and Rossby numbers $O(0.1)$ (Fig. 3a), with vorticity variance twice as large compared to winter conditions (Fig. 3c). The summer spectral slopes of the near-surface KE and salinity are both approximately $k^{-4}$ (or $k^{-3.5}$) (Fig. 3g). This deviation compared to the conventional $k^{-3}$ KE spectrum of geostrophic turbulence[51] may be caused by the Ekman spin-down of eddies due to sea ice drag and by the presence of a mixed layer, both of which will tend to steepen the subsurface KE slope. It is also possible that the model underestimates the submesoscale variability because its resolution might not be high enough to resolve a range of shear-driven instabilities forming at vorticity filaments around mesoscale eddies. Nonetheless, in striking contrast to the summer eddy field is the winter weakening of mesoscale eddies and the emergence of submesoscale variability (Fig. 3b). These heavily damped under-ice submesoscale flows have relatively small Rossby numbers of $O(0.1)$, being distinct from submesoscale flows in temperate oceans that are characterized by a $k^{-2}$ spectrum and Rossby numbers of $O(1)$[52–54]. Unlike the mesoscale, the submesoscale vorticity variance does not vary seasonally (Fig. 3c), even though in winter (i) eddies experience stronger dissipation by the sea ice and (ii) the mesoscale strain rate, which is crucial for submesoscale eddy formation via frontogenesis, is weaker by a factor of two (Supplementary Figure 5). Thus, the governing processes for

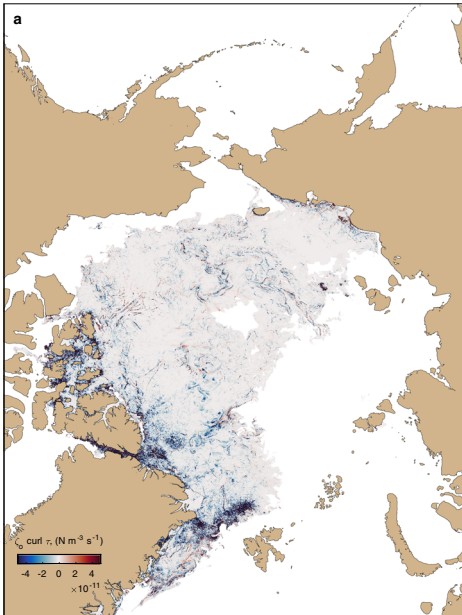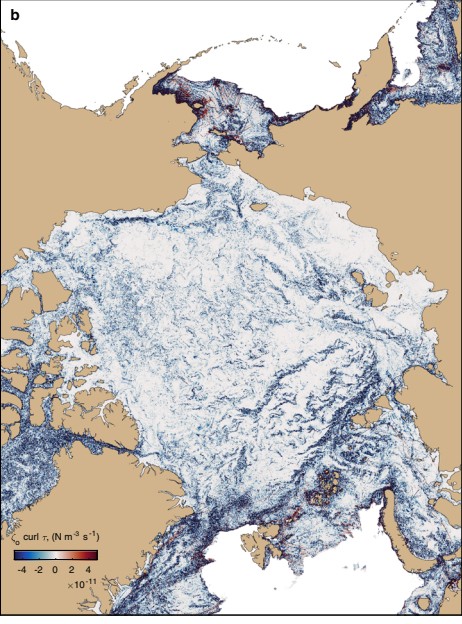

**Fig. 4 Seasonal variation in the frictional dissipation rate of the ocean vorticity variance in the ice-ocean boundary layer.** Distribution of instantaneous dissipation rate of upper-ocean (20 m depth) vorticity variance by frictional processes at the ice-ocean boundary layer ($\zeta_o \cdot curl\tau$, see Eq. (2)) plotted for (**a**) the end of summer (1st September 2012), with close to minimum sea ice extent, and (**b**) the end of winter (1st March 2012), with close to maximum sea ice extent. Color bar limits are the same in both panels with negative values corresponding to vorticity variance dissipation. The spatially-averaged dissipation rate is a factor of 6 larger in March as compared with September. Dissipation rates are enhanced at the continental shelf breaks and other regions of strong eddy kinetic energy, e.g., over the Lomonosov Ridge or in the Fram Strait.

submesoscale eddies are highly seasonal: mesoscale-driven frontogenesis dominates in the summer whereas convection or other types of mixed layer instabilities are likely more important in the winter.

From the perspective of upper-ocean currents and eddies, the heavily-packed and relatively-immobile winter sea ice can be considered as a rough lid that drains energy from upper ocean flows due to frictional dissipation via the Ekman spin-down (Equation (2)). Hence, upper-ocean variability in winter is subject to strong frictional dissipation (see Fig. 3d and 4a). Yet, in regions where sea ice concentration is relatively low, e.g., in MIZs, the ice vorticity is highly-correlated with the ocean vorticity and the frictional dissipation of upper-ocean eddies is substantially weakened (Fig. 4b). The dissipation of eddy kinetic energy (EKE) is most dramatically manifested in the mixed layer, which is relatively shallow in the Arctic Ocean, only extending ~10–20 m in the summer (Fig. 2d), but even mesoscale eddies that are centered in the interior of the water column and have surface expressions would be affected by sea ice because of the Ekman spin-down. The influence of atmospheric and sea ice processes on the ocean mixed layer is also strongly seasonal, which will further contribute to creating summer-winter differences in upper-ocean eddies and ice-ocean heat fluxes.

**Estimating eddy-heat flux feedback**. The seasonal dynamics discussed above imply a positive feedback between sea ice cover, upper-ocean eddies, and ice-ocean heat fluxes: as the sea ice melts, the key dissipation mechanism for upper-ocean variability diminishes and allows for energetic eddies to develop, bringing subsurface heat towards the sea ice, further accelerating its melt. The same feedback occurs in winter, but friction now suppresses upper-ocean eddies and leads to enhanced sea-ice growth. Assessing the strength of this feedback is challenging. The physical components of the feedback must be isolated from a myriad of other unrelated processes, and the model output only spans a single year. Nevertheless, the Arctic Ocean's spatial heterogeneity

can be used advantageously to demonstrate strong connections between the key processes in the eddy-heat flux feedback.

Regional variability in ocean-ice evolution, at seasonal time scales, is analyzed by dividing the Arctic Ocean into non-overlapping sub-domains, with 500 km sides, and diagnosing the linear sensitivity of key upper-ocean and sea-ice characteristics using monthly-mean time series averaged over the sub-domains. During the winter-spring transition, upper-ocean EKE (calculated at 20 m depth) exhibits strong sensitivity to changes in sea ice concentration; EKE becomes three times stronger following the disappearance of sea ice (Fig. 5a). The correlation between an increasing OHF and decreasing sea-ice concentration can be attributed to two factors: (i) oceanic mixed layer warming due to atmospheric warming and (ii) increased upper-ocean EKE and stirring of upper-ocean temperature. To quantify the relative importance of the latter mechanism, we introduce a linear regression model to predict the dependence of OHF on various properties, including EKE, atmosphere-ocean heat flux, sea-ice concentration, and subsurface (20 m) temperature above the freezing point. The model shows that EKE is a crucial predictor of the OHF. EKE, combined with the atmosphere-ocean heat flux, produces a correlation between predicted and diagnosed OHF of roughly 0.9; the correlation drops to 0.75 when EKE is removed as a predictor (Fig. 5b). The feedback loop is closed through the link between the sea-ice concentration tendency and the OHF. During thermodynamic growth/melt, heat transfer into the sea ice is partitioned between concentration and thickness changes, with a smaller contribution associated with heat storage. During the strong melt season (late spring to summer) and the growth season (early fall to winter), the fractional rate of change in sea ice concentration $(c_t/c)$ is significantly correlated with the fractional rate of change in sea-ice volume per unit area $(ch)_t/(ch)$, with a regression coefficient of $0.6 \pm 0.2$ (Fig. 5c). This change in sea ice volume is then directly proportional to the total heat flux (part of which are from the ocean heat fluxes) since $L\rho_i(ch)_t = -OHF + o.t.$, where $L = 333.7 \times 10^3$ J kg$^{-1}$ is the latent heat of fusion and

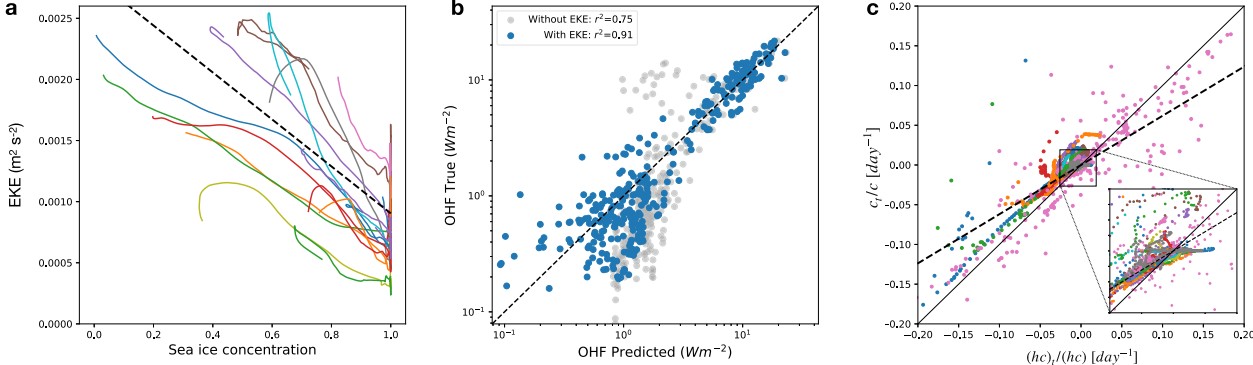

**Fig. 5 Relationships between key properties in the eddy-heat flux feedback. a** The relation between eddy kinetic energy (EKE) and sea-ice concentration; the dashed line shows the average regression based on linear regressions fits from individual sub-domains (colored curves). **b** The model-diagnosed ocean heat flux (OHF) compared to the predicted OHF using a linear regression with various predictors, including the EKE (blue) and excluding the EKE (gray). The 1:1 line is shown for reference and the $r$-squared values for gray and blue points are shown in the legend. **c** The normalized tendency in sea-ice concentration ($c_t/c$) plotted against the normalized tendency in sea-ice volume per unit area ($(hc)_t/(hc)$). The solid line is a 1:1 line and the dashed line corresponds to an average regression fit for all sub-domains with a slope of 0.6. Each line in **a**, **c** represents the temporal evolution of spatially-averaged quantities within non-overlapping 500 × 500 km sub-domains covering the entire Arctic Ocean, with a 30-day running-mean filter applied over all time series. Each point in **b** represents both the monthly and temporal mean OHF for different sub-domains.

$\rho_i = 900\,\mathrm{kg\,m^{-3}}$ is the characteristic density of sea ice; here *o.t.* represents other terms associated with atmosphere-ice and radiative heat fluxes.

Thus, a quantitative estimate of the linear sensitivity coefficients for all parts of the feedback loop follows. First, a reduced sea-ice concentration leads to an EKE increase following $EKE = -A \cdot c + o.t.$, where $A = (2 \pm 0.1) \times 10^{-3}\,\mathrm{m^2\,s^{-2}}$. Second, a linear relationship between EKE and OHF is given by $OHF = B \cdot EKE + o.t.$, where $B = 1.1 \times 10^3\,\mathrm{W\,m^{-4}\,s^2}$. Third, an increase in OHF accelerates the reduction in sea-ice concentration (as well as sea-ice thickness), following $c_t = -C \cdot OHF + o.t.$, where $C = (0.6 \pm 0.2) \cdot (L\rho_i h)^{-1} \approx 2 \times 10^{-9}\,\mathrm{m^2\,(W\,s)^{-1}}$. Combining these relationships, the evolution equation for sea-ice concentration exhibits exponential growth: $c_t = -C \cdot OHF + o.t. = -C \cdot B \cdot EKE + o.t. = +C \cdot B \cdot A \cdot c + o.t.$ Here, *o.t* denotes the combined contribution from other terms reflecting errors in the linear representation of processes that are, generally, nonlinear. Note that this feedback operates in the same way for positive concentration anomalies, leading to the amplification of sea-ice growth. We acknowledge that exponential growth in sea-ice concentration is not observed throughout the entire year due to other physical processes that provide an overall stabilizing effect. Strong external forcing can also modify the sign of anomalies depending on the season. The feedback strength described here is a product of the linear regression coefficients associated with the three stages: $A \cdot B \cdot C = 4 \times 10^{-9}\,\mathrm{s^{-1}}$. This linear feedback analysis is likely a crude representation of the full dynamics, but it provides additional physical insight consistent with the eddy-induced, ocean-ice interactions discussed in earlier sections.

The eddy-heat flux feedback will be most relevant in regions of high eddy activity with large variations in sea ice concentration, e.g., marginal or seasonal ice zones. Since this is a linear feedback, it is expected to amplify the seasonal cycle of sea growth/melt, i.e., during melt season the sea ice melts faster due to the enhanced eddy-induced heating, while during growth season the sea ice grows faster due to the suppression of eddy-induced heating. The eddy feedback may be considered weak or secondary compared to radiative feedbacks: for monthly and domain-averages, the eddy feedback modifies OHF by several W m$^{-2}$, which is small compared to the $O(100\,\mathrm{W\,m^{-2}})$ atmospheric heat fluxes. However, ocean heat fluxes occur at the ice-water interface, and

persistent OHFs as small as $O(1\,\mathrm{W\,m^{-2}})$ can affect equilibrium sea ice thickness by as much as a meter (see Fig. 9 in ref. [12]). Also, the eddy-heat flux feedback involves sea ice concentration, not its thickness. The feedback timescale, $(A \cdot B \cdot C)^{-1}$, is ~8 years, significantly longer than the seasonal cycle. Assuming that the feedback operates in MIZs over the course of a melt or growth season, the expected concentration change associated with the feedback can be crudely estimated as $\exp(0.5yr/8yr) - 1$, or 6.5%, across a single season. This change should be accounted for, especially if it leads to an expansion of the MIZ and an increase in the area over which the eddy feedback makes a significant contribution in subsequent years. Finally, these changes will likely contribute to additional feedbacks related to coupling with the atmosphere that are not captured in our relatively simple scaling.

## Discussion

In this study, we highlight the existence of a sea ice-eddy-heat flux feedback that is currently not represented in low-resolution climate models and may contribute to sea-ice prediction errors. As Arctic sea ice extent and thickness continue to decline, the key dissipation mechanisms for ocean eddies at the ice-ocean boundary layer is removed, resulting in more energetic upper-ocean mesoscale turbulence. In turn, the energized eddies enhance heat exchange between the subsurface ocean and the sea ice, accelerating its melt and constituting a positive feedback. The sea ice damping mechanism affects all eddies that have surface velocity expressions, including both mesoscale and submesoscale eddies. Thus, mesoscale-resolving Arctic Ocean models are expected to already contain the sea ice damping mechanism, and analysis of ice-eddy feedbacks in such models is highly desirable. However, we find that in winter, the OHFs are highly localized in submesoscale patches and hence models would need to be of appropriately-high resolution to represent their dynamics.

On timescales longer than seasonal, this feedback may be limited by the availability of the sub-mixed-layer heat reservoir that could be drained due to increased efficiency of ocean-ice heat exchange. Thus, it is possible that with changing Arctic sea ice characteristics, the subsurface heat will shift to depths greater than those that can be accessed by mesoscale and submesoscale processes. In this case, atmospheric-driven heating of the surface ocean would dominate sea ice growth/melt. The extent to which these limiting cases will manifest in the future Arctic depends on

details of atmosphere-ocean-ice coupling that is critically affected by both eddy stirring and small-scale turbulence. It is therefore imperative to develop and implement sea ice-aware parameterizations of upper-ocean eddies, both mesoscale and submesoscale[55], and to accurately quantify their impacts on OHFs and tracer transport in climate models.

We provided a simple but quantitative estimate of the EKE-OHF-sea ice concentration feedback by demonstrating the linear response of three key components of the feedback. The linear sensitivity analysis is based on a short model run, but sub-sampling in space permitted increased data samples. The linear sensitivity coefficients describing the EKE-concentration and the OHF-EKE relationships are unlikely to change significantly at longer timescales since these dynamics evolve over periods of several weeks to a month. Thus, the proposed feedback may impact not only seasonal sea-ice variability but also the long-term ocean-ice response to the overall trend of decreasing sea-ice concentration in the Arctic. We note that the overall role of the feedback in the seasonal development of sea ice or its decadal trends depends on the ocean and sea ice mean state. For example, models with biases towards larger fractional MIZ areas with thinner and low-concentrated sea ice are expected to be more sensitive to the eddy-heat flux feedback. Alternatively, if the model does not store sufficient heat under the mixed layer, EKE responses to changes in concentration would not necessarily be coupled to a change in the OHFs. The feedback loop may not be effectively closed in regions or times of the year where changes in sea-ice concentration are predominantly driven by external forcing, such as wind-driven sea ice advection. We thus caution that quantifying this feedback with greater certainty requires longer, high-resolution numerical experiments and, ideally, sensitivity experiments exploring sea ice rheology as this affects the critical concentration, the ice-ocean drag as this affects the eddy dissipation, and the ice-ocean heat exchange parameterization. This evidence that OHFs in the Arctic are dominated by localized filamentary and eddying structures provides a starting point for further studies on the interaction between sea ice and upper-ocean eddies. Dynamics of under-ice eddies are not well understood but, considering analogies with ice-free oceans, they should depend on such critical parameters as mixed layer depth, stratification, and strength of the mesoscale strain field—all of which are expected to change as the Arctic Ocean transitions to a more seasonal ice state. Longer-term trends in eddy dynamics in the changing Arctic Ocean are also expected to impact larger-scale circulations as well as biogeochemical cycles through lateral and vertical transport of nutrients and their impact on biomass production[56–59]. This latter topic is particularly relevant in the emerging Arctic Ocean where MIZs now cover increasingly larger areas of the Arctic.

## Methods

**Numerical model description.** Results from this study make use of output from the LLC4320 model—a high-resolution global ocean and sea ice simulation that includes tides[60]; it is a version of the Massachusetts Institute of Technology General Circulation model (MITGCM)[61]. Its grid consists of 1/48th degree horizontal spacing (~0.8 km in the Arctic Ocean) and 90 vertical levels (1 m spacing near the surface increasing towards the bottom). The model was spun-up from a consecutive set of lower-resolution configurations[62] and extended for over a year at its highest resolution, simulating a full seasonal cycle of the year 2012 with hourly output. The atmospheric forcing for the ocean-ice model comes from a 0.14 degree, 4-dimensional variational reanalysis product (European Centre for medium-Range Weather Forecasts[63]), interpolated onto the ocean grid from 6-hour time intervals. We note that the atmospheric forcing is of much larger length scales relative to the ocean resolution and hence might not provide sufficient energy to the ocean at relatively high frequencies and small length scales; in addition, since the ocean does not alter the atmosphere, a range of ocean-atmosphere-ice feedbacks might be missing. Tidal forcing was calculated off-line using a barotropic tidal model and prescribed by adding a corresponding atmospheric pressure perturbation to the ocean momentum equations; tidal forcing together with atmospheric winds are

the main sources of internal gravity waves in the model that could interact with mesoscale and submesoscale variability.

The sea ice has an active dynamic-thermodynamic model[64] with VP rheology[47]; this high-resolution simulation generates a better representation of linear kinematic features as compared to lower-resolution VP models[65]. While the continuity assumption of VP-type rheologies becomes questionable with increasing model resolution beyond characteristic floe sizes, sea ice in seasonal ice zones was found to be mostly consistent with a VP rheology[66], likely due to the smaller floe sizes as compared to pack ice. At sea ice model resolution of 1 km, particularly in the winter pack ice, we are approaching the floe scale where the sea ice behaves like a granular-type material with individual floes and anisotropic leads being present throughout. It would be ideal to use a discrete element model that could capture the granular nature of sea ice (e.g., [67–69]), but we do not yet have such modeling capability at basin scales. We can only speculate that in reality, the granular nature of sea ice floes may generate even more heterogeneous momentum and heat fluxes at the ocean-ice interface and lead to enhanced variability of mixed layer ocean currents, particularly in the submesoscale range.

Critical sea ice parameters such as the strength and lead closing constants have been adjusted in the model to ensure stability and qualitative consistency with the observed sea-ice state. However, due to the extreme computational demands, the parameters of LLC4320 model could not be optimally adjusted to represent the best match with observations, resulting in regional pattern disagreements with observations, particularly in sea ice concentration fields. In addition, at less than 1 km grid, the model only marginally permits the development of high-latitude submesoscale flows, and hence higher-order statistical quantities, such as eddy-heat and momentum fluxes or spectral energy slopes and fluxes, may be resolution-dependent. Nonetheless, the model provides a unique opportunity for fundamental dynamical process studies as it simulates critical aspects of ocean eddies[70] and sea ice dynamics[65]. Furthermore, the LLC4320 model simulates qualitatively similar winter-time submesoscale features as in idealized higher-resolution simulations[71] in a much smaller domain; and its simulated salinity spectral slope of about $k^{-3.5}$ is similar to the $k^{-3}$ slope inferred from ITP observations[72]. Thus, we have confidence that the LLC4320 model, at least partially, captures the dynamical characteristics of the upper-ocean mesoscale and submesoscale turbulence under the sea ice.

**Eddy dissipation induced by sea ice.** The frictional spin-down of geostrophic eddies by sea ice drag occurs via Ekman pumping, which is proportional to the curl of the ice-ocean stress and hence to the vorticity difference between the ocean and sea ice $w_{Ek} \sim \mathrm{curl}\,\tau_{io} \sim (\zeta_i - \zeta_o)$. The influence of sea ice drag on eddies is evident when considering the time evolution of the eddy vorticity variance, inferred by taking the curl of the mixed layer momentum equation and multiplying the result by the ocean vorticity $\zeta_o$ (see Supplementary Equation 2):

$$\frac{1}{2}\rho_o h_{b.l.} \frac{\partial \overline{\zeta_o^2}}{\partial t} = \overline{\zeta_o \cdot \mathrm{curl}\,\tau_{io}} + \text{other sources/sinks},  \quad (2)$$

where $\rho_o$ is the ocean mixed layer density and $h_{b,l}$ is the boundary layer depth below which the stress can be considered negligible. The Ekman-driven dissipation, $\zeta_o \mathrm{curl}\tau_{io}$, as diagnosed from the high-resolution numerical model LLC4320 is predominantly negative (Fig. 4). Given the magnitude of $\overline{\zeta_o \mathrm{curl}\tau_{io}} \sim O(10^{-11} - 10^{-10})$ N m$^{-2}$ s$^{-1}$ and characteristic values of $h_{b,l} \sim O(10)$ m, the sea ice drag is strong enough to dissipate upper-ocean eddies in a matter of several hours if their vorticity is confined only to the mixed layer and there were no other energy sources. However, a much longer time would be required to dissipate a subsurface mesoscale eddy because its surface vorticity expression becomes weaker under the influence of sea ice drag. From the perspective of ocean eddies, the heavily-packed sea ice could be considered a rough lid that induces dissipation proportional to eddy vorticity as $w_{Ek} \sim -\zeta_o$ in the limit of $|\zeta_i| \ll |\zeta_o|$. The magnitude of dissipation is an order of magnitude smaller in the summer because large areas of summer sea ice consist of relatively low-concentrated weakened ice with its vorticity strongly correlated to upper-ocean vorticity. It is the lack of dissipation in the summer that allows a strong surface signature of the underlying mesoscale eddy field. Note, atmospheric winds can drive the sea ice flow such that it provides a positive energy input into ocean geostrophic currents, but winds generally have a large-scale pattern, O(1000 km), and can not induce any significant sea ice vorticity at ocean eddy scales, O(10 km). Considering that ocean vorticity variance is dominated by eddies, the dissipative nature of winter sea ice is particularly intense over strong baroclinic eddying boundary currents (Fig. 4).

**Sensitivity to sea ice rheology parameters.** The critical concentration, $c_{cr}$, below which the sea ice enters the regime described by Equation (1) occurs when the ice-ocean stress is of the order of the internal sea ice stresses, i.e. $\tau \sim \nabla P(c_{cr})$ and hence must depend on the parameterization of sea ice strength $P$. We quantify the dependence of the transitional concentration on rheology by conducting a set of idealized experiments varying the parameter $C^*$ in ice pressure dependence on concentration $P = P^* ch \exp[-C^*(1 - c)]$ that is an essential ingredient of the VP parameterizaion[47,64]. We prescribe the ocean currents obeying an idealized streamfunction $\psi = \sin(kx)*\sin(ky)$ with a wavenumber $k = 2\pi/40$ km to represent the mesoscale eddy field. We then solve the sea ice equations of motion subject only

to ocean stresses; no thermodynamic forcing is applied. We calculate the dependence of the correlation $corr(\zeta_i, \zeta_o)$ on the sea ice concentration, where $\zeta_o = \nabla^2 \psi$ is the ocean vorticity and $\zeta_i$ is the equilibrium sea ice vorticity. The simulations are performed in a square, doubly-periodic domain 300 km in size and the equilibration is reached after ~1 h. The numerical model used is the MITgcm, which was configured to have the same sea ice rheology parameters as the LLC4320 simulation. The correlation and the transitional concentration are critically sensitive to the rheology parameter $C^*$: for smaller $C^*$ the critical concentration significantly decreases and the region in which sea ice is mechanically affected by ocean eddies dramatically narrows (see Supplementary Section 4 and figures therein). For the most commonly used value of parameter $C^* = 20$, the transition between ice regimes occurs at a concentration of ~80% (Supplementary Figure 3), which also defines the width of the region over which the eddy dissipation is negligible. We note, however, that the presence of different ice types will affect $C^*$, which is kept constant in VP rheology. Improving sea ice rheology and validating it with observations remains a significant challenge in sea ice modeling.

## Data availability

The output from the high-resolution ocean model simulation (LLC4320) is available at the NASA ECCO Data Portal https://data.nas.nasa.gov/.

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

## Acknowledgements
The authors gratefully acknowledge support from the following funding sources: NSF grant OCE-1829969, G.E.M. and A.F.T.; ONR grant N00014-19-1-2421, G.E.M., and A.F.T.; the Stanback Postdoctoral Fellowship Fund, G.E.M.; the Davidow Discovery Fund, G.E.M.; and the Terrestrial Hazard Observations and Reporting program at Caltech, A.F.T. The manuscript benefited from discussions at the annual Forum for Arctic modeling and Observing Synthesis (FAMOS) funded by the NSF OPP awards PLR-1313614 and PLR-1203720. The authors acknowledge the high-performance computing support from JPL/NASA and D. menemenlis for conducting the LLC4320 high-resolution numerical simulation. The authors thank Robert Fajber for discussions of the open-water area formation efficiency for sea ice models as well as comments from three reviewers that improved the manuscript.

## Author contributions
G.E.M. conceived the study and together with A.F.T. analyzed the data from the high-resolution simulation and idealized sensitivity experiments. Both authors wrote the manuscript.

## Competing interests
The authors declare no competing interests.
