## [Peer Review File · Nature Communications]

Heavy footprints of upper-ocean eddies on weakened Arctic sea ice in marginal ice zonesReviewers' Comments:

Reviewer #1:

Remarks to the Author:

Overall comments:

This is an interesting novel manuscript of wide significance for polar oceanographers and climate modelers. It proposes a potentially important positive feedback mechanism of Arctic Ocean eddies on sea ice. The feedback might be important for improving coupled climate model projections of Arctic sea ice decline. The paper is well-written with high quality figures. It may be suitable for publication in Nature Communications, but there are several major issues that need resolving first, plus many specific (mainly minor) issues. Overall, I recommend a Major Revision.

General comments:

1. The regime switch from uncorrelated to correlated surface ocean and sea ice vorticity moving through the marginal ice zone (or through the seasonal cycle) is an important result (Figure 1). The scaling in (1) is also useful. However, the mechanism of the regime transition relies on the critical sea ice concentration $c_{cr} \sim 0.8$. For sea ice concentrations c below c_{cr} , the sea ice acts passively without internal stresses and the sea ice vorticity converges to the surface ocean vorticity. The critical sea ice concentration c_{cr} is therefore a key parameter. The emergence of $c_{cr} \sim 0.8$ is itself based on the parameterization of sea ice mechanics, however, which are not fully understood. Specifically, the viscous-plastic rheology says ice strength scales as $c \exp(-C^* (1-c))$ (see page 4 and supplementary information page 3). I wonder about the realism of this sea ice strength parameterization (other schemes exist). The main findings of the paper (Figs. 1–4) rely on c_{cr} and the regime transition it delineates. The findings are plausible, but the reader needs to understand that they're built on a (somewhat) uncertain parametrization of sea ice mechanics.

2. For these reasons, among others, I think the paper needs a careful discussion of observed Arctic ocean eddies. It's conspicuous that almost no observations are included (Figure S1 and its text are too superficial). In fact, much is known. While a detailed quantitative comparison with the model is not needed, the reader should have a clear idea about the characteristics of natural eddies and any known systematic model biases.

3a. The authors propose a new positive feedback mechanism whereby sea ice cover, upper-ocean mixing and ocean-ice heat fluxes interact. At high concentrations (e.g. in winter), sea ice effectively dissipates (damps) upper ocean mesoscale eddies via ice-ocean stresses. The ocean to ice heat flux, which the eddies mediate, is therefore low. At low concentrations (e.g., in summer and in the marginal ice zone), the upper ocean mesoscale eddies are more active and provide a larger heat flux. The effect on the sea ice cover is therefore to promote high concentrations and diminish low concentrations. This is an interesting idea. No direct evidence of the feedback mechanism is presented, and its main attributes (like feedback timescale) aren't quantified. The paper would be strengthened if it contained evidence of and details on this feedback mechanism. At least, the paper should articulate the next research questions on this issue.

3b. It's also worth noting that the feedback mechanism involves sea ice damping of mesoscale ($\sim 30\text{km}$) anticyclonic eddies, not submesoscale eddies (or cyclones). That, presumably, implies that a mesoscale (but not submesoscale) eddy-resolving model would include the feedback. Some discussion is needed.

Specific comments:

The paper repeatedly talks about Arctic sea ice "melt" being responsible for the decline in summer-

time ice cover. This needs to be corrected. Although greater melt may be important, declining sea ice formation (freezing) is also important, perhaps dominant. If there's less freezing, sea ice declines without any change in melt.

There are many reasons that climate model projections (typically) underestimate the decline in sea ice over the last 30 years. Biases in ocean heat fluxes to the sea ice are one explanation (Abstract, 1st paragraph of subsection Main, and elsewhere), but there are others. A more balanced discussion is needed of possible mechanisms responsible for climate models underestimating sea ice decline and the related Arctic amplification. E.g., See Pithan and Mauritsen (2014, Nature Geo.) and Graverson et al. (2014, J. Climate).

Abstract: The sentence starting "Additionally, under ice eddy energetics..." is unclear and needs to be reworded.

Main, 1st paragraph. It says: "Climate models make inaccurate sea ice predictions beyond several months..." This statement needs to be reworded or clarified because it's not literally true (I don't think climate models don't make seasonal predictions of sea ice). But I think what's meant is that the observed sea ice decline is underestimated by many climate projections, which is true. But it's not universally true. In fact, several climate models match the observed decline in summer sea ice extent. E.g., see Wang and Overland (2009, GRL) and Liu et al. (2013, PNAS).

Main, 2nd paragraph. It says: "Under-ice Arctic Ocean turbulence is far from quiescent..." yet it says later (2nd page) that there is "relatively weak vertical mixing" in the Arctic Ocean. Be specific. In fact, a great range of small-scale 3D turbulence intensities exists, including some of the least energetic regimes in the global ocean. E.g., see Fer (2009, Atmos. Ocean. Sci. Lett.) and Rudels, Anderson and Jones (1996, JGR).

Main, 3rd paragraph. It says that ocean sub grid scale mixing parametrizations have only "been developed for ice-free oceans". And I agree that "sea-ice aware parametrizations of upper-ocean eddies" (page 3) are needed. Yet, there are some papers on the topic of under-ice parametrizations, such as Ramudu et al. (2017, JGR), Skillingstad and Denbo (2001, JGR) and Horvat et al. (2016). A more balanced discussion is needed.

Strong Mechanical...3rd paragraph: The text on critical concentration c_{cr} is confusing here and too compressed.

Figure 2: What sets the value (about 0.7) of the sea ice and upper ocean vorticity correlation below 80% concentrations (panel a)? Why is there a tail to large heat fluxes for cyclones (panel d)? The units (m) of "sea ice volume anomaly" in panel c look wrong. The evidence for sea ice accumulation in cyclones and repulsion in anticyclones (panel c) isn't very strong. What's the statistical significance of the difference between these pdfs? This "mechanical redistribution of sea ice towards cyclones" is argued to be an important process in this paper. Maybe condition the pdfs on low winds and replot?

Enhancement of ... heat fluxes: 1st paragraph: It says "the locally enhanced heating over anticyclones will influence a larger sea ice area..." Larger than what? Also, anticyclones tend to repel sea ice, at least for loosely packed floes. That reduces the heating-enhancement effect. What's the relative importance of these tendencies? Also, how is this related to the observed domination of mesoscale anticyclones over cyclones (at least in the Beaufort Gyre)? See Zhao et al. (2014, JGR).

Figure 3: What is the ice cover for the summer and winter snapshots shown? Based on the main text, we should expect ocean to ice heat flux anomalies concentrated in anticyclones. How is this evident in Fig. 3? Some discussion is needed. Why are the y-axes on panels c and d different and what are the units?

Seasonality... 2nd paragraph: It says about submesoscale vorticity variance that "frontogenesis dominates in the summer and convection-driven instabilities in the winter" What is the evidence to support this statement?

Figure 4: State the units of dissipation rate.

Discussion: 1st paragraph: It talks about "below the effective depth of submesoscale penetration" switching off the positive feedback mechanism. But the feedback involves mesoscale eddies, not submesoscale ones. This is unclear. Also, be specific when using the word "turbulence" (used in many places). What, precisely, do you mean?

Discussion: Final paragraph: I found this text disconnected and superfluous. I think it could be cut.

Methods:

Eddy dissipation: It says the vorticity variance equation (2) comes from the supplementary eq. (2), but I don't see the connection.

Supplementary Information:

Section 1: It says "The characteristic spatial scales of 10-50 km and persistence times of several days point to coupled dynamics with ocean eddies." Why? It's plausible that the dynamics are coupled, but this seems like weak evidence to me. Also, Fig S1 plausibly shows sea ice "localized in small-scale eddies, filaments, fronts..." but again, the evidence is weak. Without independent knowledge of the surface ocean current (e.g. its vorticity), why do you discount other possibilities? Also the text talks about the Labrador Current, which is not seen in the figure and not about the East Siberian shelf, which is.

Section 3:

This is a long technical section. Several issues need fixing and careful revision. The focus and precision need improvement.

Why is there no wind forcing in equations (1) to (4)? Define m_o . Why isn't there an equation for evolution of thickness h ? (such an h eqn would replace (3), which seems to assume h is a uniform constant). The sea ice momentum is neglected (which is reasonable); what about the ocean momentum (which is much larger than sea ice momentum according to the text)?

The sentence "Even without the rheology terms...i.e., the classical Ekman balance." is confusing. Reword. Also, explain why cyclones "lead to sea ice accumulation, and anticyclones repel the ice".

In what sense are (13) and (14) "solved in the MITGCM"? (what are the dependent and independent variables?). Why is this discussion necessary?

The text between (14) and (15) says that if the ocean and ice speeds are different, the concentration c must vary with x . But what about h varying with x (or P^* , which doesn't seem to be defined).

Why is the Rossby number defined after (15)? It doesn't seem to be used.

Explain/justify the statement "the sea ice adjustment time scale of several hours".

"The scaling law dictates how close the ocean and ice vorticities are allowed to be in viscous regime where the sea ice is mobile" doesn't make sense to me. Reword/clarify.

The final summary paragraph doesn't do a good job of wrapping up the section. Many readers will have lost the thread by this point. The final comment about "the sea ice strength variable P decrease to zero across the transition..." is already guaranteed by the expression for P below eq (9). Isn't the main point eq (5) and the scaling (15)? [Although the argument seems to set $h = \text{constant}$, so (15) is ambiguous to interpret].

There are several typos in this section. E.g., m should be m_i , ρ should be ρ_0 some places (or maybe ρ_i), "Raynolds", "homogeneity along the x direction" (should read "y direction"?), $\epsilon_{\{i,j\}}$ in (12) is missing a dot above it (?). What is ρ_w ? What is "strongly non-linear"? Fix "onc"

Section 4:

Why aren't C^* values larger than 20 explored?

Why is the regime transition near $(c=c_{cr}, C^*) \sim (0.8, 20)$? That point lies in the asymptotic regime of $P \sim c \exp(-C^*(1-c))$ where the argument to the exponent function is ~ -4 , which surprises me a bit.

Reviewer #2:

Remarks to the Author:

Review of 'Heavy footprints of upper ocean eddies on weakened Arctic sea ice in marginal ice zones' by Manucharyan & Thompson, submitted to Nature Communication.

In this manuscript, the authors analyze outputs from a global model run at extremely high resolution ($\sim 1\text{km}$) over a year, in order to seek evidence of the signature of ocean turbulence (at meso and submesoscales) onto sea ice. The topic is highly relevant as the results are undoubtedly providing us with a new process by which the ocean might have contributed to the on-going sea ice retreat (and will contribute in the future). The results are also questioning how much predictions on the future sea ice decline (which affects many features of the climate) obtained by state of the art climate models are reliable, given that the eddy-sea ice interactions are not represented nor parameterized in those models. In this regard, the findings of the paper are an extremely useful contribution for the community at large.

Yet, I have to admit that I am really puzzled and I am not sure of what to think of the paper as it is now. Although I am keen to believe that the science is sound, I find that the results are very poorly supported by the figures and the diagnostics, the details of which are extremely hard to follow. The analysis is somewhere between being extremely quantitative (which is made possible by the model) and extremely qualitative, with rough numbers given without any supporting evidence. This is very frustrating, and undermine greatly the impact that the paper could have in the community. I do not think that the paper can be published in nature communication as it is, and I would recommend that the authors take time to make an in-depth analysis of the different features they examine here.

In the following, I'll go through the manuscript from the beginning, noting issues as they appear.

Main

The part in paragraph 3 describing how sea ice can affect ocean turbulence is in reality the results of the present study. It is thus weird to have it here in the middle of the context and the literature review.

Strong mechanical sea ice-ocean coupling in MIZs

- This first part of this section is mainly supported by Figure 1, which shows a snapshot in July 2012. How is that snapshot representative of 'a melt season'?
- Looking at figure 1 (and the movie in Supp. Material) it is clear that the sea ice is heavily biased in

the simulation, with a whole in the middle of the basin, and a MIZ that is way too big. Given this bias and the fact that the strong coupling between the ocean turbulence and sea ice occurs for concentration lower than 80%, is it possible that in reality, such a coupling would only occur in a very small part of the basin? This questions the fact that this coupling would actually be of any importance for the dynamics of the sea ice in the real Arctic.

- The authors should be more upfront and at the very least discuss this point. Some quantification of something like the surface and the time (over the seasonal cycle) of where and when the coupling is important would also help. The movie is really nice, but does not provide this quantification (and dates should be included in the movie)
- Given that it seems that it is sea ice concentration that is the critical factor for the ocean-sea ice coupling, why the authors are not showing a map of it instead of the concentration in Figure 1?
- Moreover, in panel C, concentration is shown but is not linked to any axis. The x-axis is also not linked to anything (I guess it is distance along the line in panel b, but the numbers seem arbitrary).
- What is 'upper ocean'? Is it really just the top layer of the model? Is the coupling occurring only with the mixed layer turbulence or the sea ice vorticity is also affected by the turbulence below the mixed layer? This is never discussed in the text while I think this is an important aspect. The depth of the mixed layer should be given too somewhere.
- The last part in this section and Figure 2c is not convincing. Again, the authors show one given eddy to prove their point, but we have no idea how much this specific eddy is representative any 'normal' state. The lack of difference between the two distributions makes it very hard to believe that there is indeed a significant difference between cyclone and anticyclones.
- More generally, the caption does not allow us to fully understand what is done. Is the distribution computed from the one snapshot shown on Fig.1? And if that's the case, again, how much can this be seen a representative of the Arctic Summer?

Enhancement of ocean-ice heat fluxes by eddies

- This section provides some kind of quantitative results but, again, we do not know how they were obtained. How is the '80%' obtained? Note that the same sentence is repeated in the following section. How the heat flux over eddy and outside are compared? ^{SEP} The authors talk about 'domain averaged' heat flux but no number is given, and no figure is shown. We are again somewhere in the grey zone where nothing is fully quantitative in the analysis but the text does not really reflect this.

Seasonality of heat fluxes, upper ocean turbulence and eddy dissipation by sea ice

- the first paragraph is mostly a repetition of the previous paragraph, and again many statements are not supported by any evidence (80% of the heat flux, 10% of the area...)
- What are we looking at in Figure 3? We do not even know when and where the boxes are. Given that 2 regimes are identified in section 1 depending on the concentration, are those diagnostics depends on the regime considered (and is which regime are we here?)
- The part about the under ice dissipation is interesting but again, not fully supported by evidences. How do the authors know that the generation of submesoscale differ in winter and summer?
- Given that the model has run over a full year, I do not understand why the authors are not presenting here a full seasonal cycle of some key diagnostics (e.g. heat flux, some kind of integrated quantity representing the ocean turbulence...)

Methods

- The part about the rheology should be moved to supplementary material given that it is not directly relevant for the core of the manuscript. Some basic evaluation of the key features that matter for the present study (e.g. sea ice concentration, mixed layer depth) should be shown
- The atmospheric forcing used for the simulation is very low resolution compared to resolution of the model. I am not questioning this choice, but how much would you expect that using a forcing at higher resolution could results in small scale sea ice features being also forced by the atmosphere? This should be discussed.

Reviewer #3:

Remarks to the Author:

This paper presents new and important work that may have real consequences for sea ice prediction. It may be a factor in the "faster than forecast" problem that helps explain why sea ice in climate models have generally not declined as fast as is observed. The authors present a detailed analysis and series of sensitivity experiments that effectively support their conclusions. I am in favor of seeing this kind of rigorous modeling/theoretical work published in Nature Climate Change as it has relevance to predicting future change. I believe it will be of interest to all who work on predicting climate change and especially those who are concerned with sea ice.

I have a few questions for further thought and some minor comments.

General points

Please add line numbers to your papers in future submissions. The figures need to be readable by the color blind as well.

The mean state of the sea ice in the model needs to be discussed. It appears to be biased thin, which would possibly place the sea ice state below the critical value for τ . I suppose it is thin because of this positive feedback whereby the ocean is stirring up more heat to the sea ice. It appears to be too much though.

The authors emphasize the role of sea ice concentration as the control on the frictional dissipation via the sea ice strength. However, the thickness is normally an even larger control. But the supplemental explains that $P \sim c \exp[-C*(1-c)]$ where as in Hibler 79, $P \sim h \exp[-C*(1-c)]$. c and h are not linearly related year-round so it is not reasonable to simply swap one for the other. Was the strength parameterization actually coded with c in place of h ? And if so, what if it were not? How would Figure S3 look compared to h rather than c ? Isn't h even more important? Is the critical concentration really a critical thickness?

Have ocean-ice heat fluxes of 30 W/m² ever been measured in the central Arctic where the sea ice concentration is ~80%? Has anyone ever found that heat fluxes are larger than average in anticyclones. Presumably the authors meant an anticyclonic ocean eddy, but it was not my first assumption. Please note whether the circulations mentioned are in the ocean, ice or atmosphere.

A paper that is relevant to this one is by David Holland in J. Climate 2001 "An Impact of Subgrid-Scale Ice-Ocean Dynamics on Sea-Ice Cover"

Specific points in the Main section

I believe the first paper to discuss the speed up in sea ice with thinning is Rampal et al (2011). They frame the relationship as a positive feedback and therefore should be reviewed and cited.

"On interannual and longer timescales, climate projection models tend to overestimate sea ice extent I don't understand what timescale has to do with a the claimed bias. I also don't think it is true that models tend to overestimate sea ice extent. There are many that underestimate it, such as the GISS model in CMIP5.

What is meant by "remain largely unconstrained on basin scales" in paragraph one? Most things are unconstrained in climate simulations since there is no data assimilation and the initial conditions are long since forgotten.

Please add "Nearly always" to "Climate models do not resolve ice-ocean interactions at eddy scales" in

paragraph two. An exception is PRIMAVERA with FESOM.

"sea distributions"? presumably the word "ice" is missing.

The statement "enhancement in the correlation between the ice and ocean vorticity in MIZs" needs to say what the enhancement is relative to.

Specific points in the Methods section

What is the resolution of the atmospheric forcing and how does it influence the simulation to have it so comparably smooth?

The authors argue that VP has been found to be appropriate for the seasonal ice zones, but then they analyze their model in the perennial zones. What are the consequences of the VP assumption at such fine resolution where the model resolution is smaller than the floe size?

Something is wrong with this the english in this sentence: "the model only marginally permitting the development of submesoscale flows...". I think permitting should be permits.

Fig 2c why are units of volume in m? And is the inset an anomaly? How can the thickness change so rapidly during a cyclone? By what mechanism?

Fig 2 caption "correspondingly" is used where "respectively" is meant. The xlabel on the panels came out strangely.

Fig 4 fix the aspect ratios of the map projections and order winter and summer the same as in Fig 3.

Reviewers' Comments:

Reviewer #1:

Remarks to the Author:

Overall comments:

This is an interesting novel manuscript of wide significance for polar oceanographers and climate modelers. It proposes a potentially important positive feedback mechanism of Arctic Ocean eddies on sea ice. The feedback might be important for improving coupled climate model projections of Arctic sea ice decline. The paper is well-written with high quality figures. It may be suitable for publication in Nature Communications, but there are several major issues that need resolving first, plus many specific (mainly minor) issues. Overall, I recommend a Major Revision.

General comments:

1. The regime switch from uncorrelated to correlated surface ocean and sea ice vorticity moving through the marginal ice zone (or through the seasonal cycle) is an important result (Figure 1). The scaling in (1) is also useful. However, the mechanism of the regime transition relies on the critical sea ice concentration $c_{cr} \sim 0.8$. For sea ice concentrations c below c_{cr} , the sea ice acts passively without internal stresses and the sea ice vorticity converges to the surface ocean vorticity. The critical sea ice concentration c_{cr} is therefore a key parameter. The emergence of $c_{cr} \sim 0.8$ is itself based on the parameterization of sea ice mechanics, however, which are not fully understood. Specifically, the viscous-plastic rheology says ice strength scales as $c \exp(-C^* (1-c))$ (see page 4 and supplementary information page 3). I wonder about the realism of this sea ice strength parameterization (other schemes exist). The main findings of the paper (Figs. 1–4) rely on c_{cr} and the regime transition it delineates. The findings are plausible, but the reader needs to understand that they're built on a (somewhat) uncertain parametrization of sea ice mechanics.

2. For these reasons, among others, I think the paper needs a careful discussion of observed Arctic ocean eddies. It's conspicuous that almost no observations are included (Figure S1 and its text are too superficial). In fact, much is known. While a detailed quantitative comparison with the model is not needed, the reader should have a clear idea about the characteristics of natural eddies and any known systematic model biases.

3a. The authors propose a new positive feedback mechanism whereby sea ice cover, upper-ocean mixing and ocean-ice heat fluxes interact. At high concentrations (e.g. in winter), sea ice effectively dissipates (damps) upper ocean mesoscale eddies via ice-ocean stresses. The ocean to ice heat flux, which the eddies mediate, is therefore low. At low concentrations (e.g., in summer and in the marginal ice zone), the upper ocean mesoscale eddies are more active and provide a larger heat flux. The effect on the sea ice cover is therefore to promote high concentrations and diminish low concentrations. This is an interesting idea. No direct evidence of the feedback mechanism is presented, and its main attributes (like feedback timescale) aren't quantified. The paper would be strengthened if it contained evidence of and details on this feedback mechanism. At least, the paper should articulate the next research questions on this issue.

3b. It's also worth noting that the feedback mechanism involves sea ice damping of mesoscale (~ 30 km) anticyclonic eddies, not submesoscale eddies (or cyclones). That, presumably, implies that a mesoscale (but not submesoscale) eddy-resolving model would include the feedback. Some discussion is needed.

Specific comments:

The paper repeatedly talks about Arctic sea ice "melt" being responsible for the decline in summer-

time ice cover. This needs to be corrected. Although greater melt may be important, declining sea ice formation (freezing) is also important, perhaps dominant. If there's less freezing, sea ice declines without any change in melt.

There are many reasons that climate model projections (typically) underestimate the decline in sea ice over the last 30 years. Biases in ocean heat fluxes to the sea ice are one explanation (Abstract, 1st paragraph of subsection Main, and elsewhere), but there are others. A more balanced discussion is needed of possible mechanisms responsible for climate models underestimating sea ice decline and the related Arctic amplification. E.g., See Pithan and Mauritsen (2014, Nature Geo.) and Graverson et al. (2014, J. Climate).

Abstract: The sentence starting "Additionally, under ice eddy energetics..." is unclear and needs to be reworded.

Main, 1st paragraph. It says: "Climate models make inaccurate sea ice predictions beyond several months..." This statement needs to be reworded or clarified because it's not literally true (I don't think climate models don't make seasonal predictions of sea ice). But I think what's meant is that the observed sea ice decline is underestimated by many climate projections, which is true. But it's not universally true. In fact, several climate models match the observed decline in summer sea ice extent. E.g., see Wang and Overland (2009, GRL) and Liu et al. (2013, PNAS).

Main, 2nd paragraph. It says: "Under-ice Arctic Ocean turbulence is far from quiescent..." yet it says later (2nd page) that there is "relatively weak vertical mixing" in the Arctic Ocean. Be specific. In fact, a great range of small-scale 3D turbulence intensities exists, including some of the least energetic regimes in the global ocean. E.g., see Fer (2009, Atmos. Ocean. Sci. Lett.) and Rudels, Anderson and Jones (1996, JGR).

Main, 3rd paragraph. It says that ocean sub grid scale mixing parametrizations have only "been developed for ice-free oceans". And I agree that "sea-ice aware parametrizations of upper-ocean eddies" (page 3) are needed. Yet, there are some papers on the topic of under-ice parametrizations, such as Ramudu et al. (2017, JGR), Skillingstad and Denbo (2001, JGR) and Horvat et al. (2016). A more balanced discussion is needed.

Strong Mechanical...3rd paragraph: The text on critical concentration c_{cr} is confusing here and too compressed.

Figure 2: What sets the value (about 0.7) of the sea ice and upper ocean vorticity correlation below 80% concentrations (panel a)? Why is there a tail to large heat fluxes for cyclones (panel d)? The units (m) of "sea ice volume anomaly" in panel c look wrong. The evidence for sea ice accumulation in cyclones and repulsion in anticyclones (panel c) isn't very strong. What's the statistical significance of the difference between these pdfs? This "mechanical redistribution of sea ice towards cyclones" is argued to be an important process in this paper. Maybe condition the pdfs on low winds and replot?

Enhancement of ... heat fluxes: 1st paragraph: It says "the locally enhanced heating over anticyclones will influence a larger sea ice area..." Larger than what? Also, anticyclones tend to repel sea ice, at least for loosely packed floes. That reduces the heating-enhancement effect. What's the relative importance of these tendencies? Also, how is this related to the observed domination of mesoscale anticyclones over cyclones (at least in the Beaufort Gyre)? See Zhao et al. (2014, JGR).

Figure 3: What is the ice cover for the summer and winter snapshots shown? Based on the main text, we should expect ocean to ice heat flux anomalies concentrated in anticyclones. How is this evident in Fig. 3? Some discussion is needed. Why are the y-axes on panels c and d different and what are the units?

Seasonality... 2nd paragraph: It says about submesoscale vorticity variance that "frontogenesis dominates in the summer and convection-driven instabilities in the winter" What is the evidence to support this statement?

Figure 4: State the units of dissipation rate.

Discussion: 1st paragraph: It talks about "below the effective depth of submesoscale penetration" switching off the positive feedback mechanism. But the feedback involves mesoscale eddies, not submesoscale ones. This is unclear. Also, be specific when using the word "turbulence" (used in many places). What, precisely, do you mean?

Discussion: Final paragraph: I found this text disconnected and superfluous. I think it could be cut.

Methods:

Eddy dissipation: It says the vorticity variance equation (2) comes from the supplementary eq. (2), but I don't see the connection.

Supplementary Information:

Section 1: It says "The characteristic spatial scales of 10-50 km and persistence times of several days point to coupled dynamics with ocean eddies." Why? It's plausible that the dynamics are coupled, but this seems like weak evidence to me. Also, Fig S1 plausibly shows sea ice "localized in small-scale eddies, filaments, fronts..." but again, the evidence is weak. Without independent knowledge of the surface ocean current (e.g. its vorticity), why do you discount other possibilities? Also the text talks about the Labrador Current, which is not seen in the figure and not about the East Siberian shelf, which is.

Section 3:

This is a long technical section. Several issues need fixing and careful revision. The focus and precision need improvement.

Why is there no wind forcing in equations (1) to (4)? Define m_o . Why isn't there an equation for evolution of thickness h ? (such an h eqn would replace (3), which seems to assume h is a uniform constant). The sea ice momentum is neglected (which is reasonable); what about the ocean momentum (which is much larger than sea ice momentum according to the text)?

The sentence "Even without the rheology terms...i.e., the classical Ekman balance." is confusing. Reword. Also, explain why cyclones "lead to sea ice accumulation, and anticyclones repel the ice".

In what sense are (13) and (14) "solved in the MITGCM"? (what are the dependent and independent variables?). Why is this discussion necessary?

The text between (14) and (15) says that if the ocean and ice speeds are different, the concentration c must vary with x . But what about h varying with x (or P^* , which doesn't seem to be defined).

Why is the Rossby number defined after (15)? It doesn't seem to be used.

Explain/justify the statement "the sea ice adjustment time scale of several hours".

"The scaling law dictates how close the ocean and ice vorticities are allowed to be in viscous regime where the sea ice is mobile" doesn't make sense to me. Reword/clarify.

The final summary paragraph doesn't do a good job of wrapping up the section. Many readers will have lost the thread by this point. The final comment about "the sea ice strength variable P decrease to zero across the transition..." is already guaranteed by the expression for P below eq (9). Isn't the main point eq (5) and the scaling (15)? [Although the argument seems to set $h = \text{constant}$, so (15) is ambiguous to interpret].

There are several typos in this section. E.g., m should be m_i , ρ should be ρ_0 some places (or maybe ρ_i), "Raynolds", "homogeneity along the x direction" (should read "y direction"?), $\epsilon_{\{i,j\}}$ in (12) is missing a dot above it (?). What is ρ_w ? What is "strongly non-linear"? Fix "onc"

Section 4:

Why aren't C^* values larger than 20 explored?

Why is the regime transition near $(c=c_{cr}, C^*) \sim (0.8, 20)$? That point lies in the asymptotic regime of $P \sim c \exp(-C^* (1-c))$ where the argument to the exponent function is ~ -4 , which surprises me a bit.

Reviewer #2:

Remarks to the Author:

Review of 'Heavy footprints of upper ocean eddies on weakened Arctic sea ice in marginal ice zones' by Manucharyan & Thompson, submitted to Nature Communication.

In this manuscript, the authors analyze outputs from a global model run at extremely high resolution ($\sim 1\text{km}$) over a year, in order to seek evidence of the signature of ocean turbulence (at meso and submesoscales) onto sea ice. The topic is highly relevant as the results are undoubtedly providing us with a new process by which the ocean might have contributed to the on-going sea ice retreat (and will contribute in the future). The results are also questioning how much predictions on the future sea ice decline (which affects many features of the climate) obtained by state of the art climate models are reliable, given that the eddy-sea ice interactions are not represented nor parameterized in those models. In this regard, the findings of the paper are an extremely useful contribution for the community at large.

Yet, I have to admit that I am really puzzled and I am not sure of what to think of the paper as it is now. Although I am keen to believe that the science is sound, I find that the results are very poorly supported by the figures and the diagnostics, the details of which are extremely hard to follow. The analysis is somewhere between being extremely quantitative (which is made possible by the model) and extremely qualitative, with rough numbers given without any supporting evidence. This is very frustrating, and undermine greatly the impact that the paper could have in the community. I do not think that the paper can be published in nature communication as it is, and I would recommend that the authors take time to make an in-depth analysis of the different features they examine here.

In the following, I'll go through the manuscript from the beginning, noting issues as they appear.

Main

The part in paragraph 3 describing how sea ice can affect ocean turbulence is in reality the results of the present study. It is thus weird to have it here in the middle of the context and the literature review.

Strong mechanical sea ice-ocean coupling in MIZs

- This first part of this section is mainly supported by Figure 1, which shows a snapshot in July 2012. How is that snapshot representative of 'a melt season'?
- Looking at figure 1 (and the movie in Supp. Material) it is clear that the sea ice is heavily biased in

the simulation, with a whole in the middle of the basin, and a MIZ that is way too big. Given this bias and the fact that the strong coupling between the ocean turbulence and sea ice occurs for concentration lower than 80%, is it possible that in reality, such a coupling would only occur in a very small part of the basin? This questions the fact that this coupling would actually be of any importance for the dynamics of the sea ice in the real Arctic.

- The authors should be more upfront and at the very least discuss this point. Some quantification of something like the surface and the time (over the seasonal cycle) of where and when the coupling is important would also help. The movie is really nice, but does not provide this quantification (and dates should be included in the movie)
- Given that it seems that it is sea ice concentration that is the critical factor for the ocean-sea ice coupling, why the authors are not showing a map of it instead of the concentration in Figure 1?
- Moreover, in panel C, concentration is shown but is not linked to any axis. The x-axis is also not linked to anything (I guess it is distance along the line in panel b, but the numbers seem arbitrary).
- What is 'upper ocean'? Is it really just the top layer of the model? Is the coupling occurring only with the mixed layer turbulence or the sea ice vorticity is also affected by the turbulence below the mixed layer? This is never discussed in the text while I think this is an important aspect. The depth of the mixed layer should be given too somewhere.
- The last part in this section and Figure 2c is not convincing. Again, the authors show one given eddy to prove their point, but we have no idea how much this specific eddy is representative any 'normal' state. The lack of difference between the two distributions makes it very hard to believe that there is indeed a significant difference between cyclone and anticyclones.
- More generally, the caption does not allow us to fully understand what is done. Is the distribution computed from the one snapshot shown on Fig.1? And if that's the case, again, how much can this be seen a representative of the Arctic Summer?

Enhancement of ocean-ice heat fluxes by eddies

- This section provides some kind of quantitative results but, again, we do not know how they were obtained. How is the '80%' obtained? Note that the same sentence is repeated in the following section. How the heat flux over eddy and outside are compared? ^{SEP} The authors talk about 'domain averaged' heat flux but no number is given, and no figure is shown. We are again somewhere in the grey zone where nothing is fully quantitative in the analysis but the text does not really reflect this.

Seasonality of heat fluxes, upper ocean turbulence and eddy dissipation by sea ice

- the first paragraph is mostly a repetition of the previous paragraph, and again many statements are not supported by any evidence (80% of the heat flux, 10% of the area...)
- What are we looking at in Figure 3? We do not even know when and where the boxes are. Given that 2 regimes are identified in section 1 depending on the concentration, are those diagnostics depends on the regime considered (and is which regime are we here?)
- The part about the under ice dissipation is interesting but again, not fully supported by evidences. How do the authors know that the generation of submesoscale differ in winter and summer?
- Given that the model has run over a full year, I do not understand why the authors are not presenting here a full seasonal cycle of some key diagnostics (e.g. heat flux, some kind of integrated quantity representing the ocean turbulence...)

Methods

- The part about the rheology should be moved to supplementary material given that it is not directly relevant for the core of the manuscript. Some basic evaluation of the key features that matter for the present study (e.g. sea ice concentration, mixed layer depth) should be shown
- The atmospheric forcing used for the simulation is very low resolution compared to resolution of the model. I am not questioning this choice, but how much would you expect that using a forcing at higher resolution could results in small scale sea ice features being also forced by the atmosphere? This should be discussed.

Reviewer #3:

Remarks to the Author:

This paper presents new and important work that may have real consequences for sea ice prediction. It may be a factor in the "faster than forecast" problem that helps explain why sea ice in climate models have generally not declined as fast as is observed. The authors present a detailed analysis and series of sensitivity experiments that effectively support their conclusions. I am in favor of seeing this kind of rigorous modeling/theoretical work published in Nature Climate Change as it has relevance to predicting future change. I believe it will be of interest to all who work on predicting climate change and especially those who are concerned with sea ice.

I have a few questions for further thought and some minor comments.

General points

Please add line numbers to your papers in future submissions. The figures need to be readable by the color blind as well.

The mean state of the sea ice in the model needs to be discussed. It appears to be biased thin, which would possibly place the sea ice state below the critical value for τ . I suppose it is thin because of this positive feedback whereby the ocean is stirring up more heat to the sea ice. It appears to be too much though.

The authors emphasize the role of sea ice concentration as the control on the frictional dissipation via the sea ice strength. However, the thickness is normally an even larger control. But the supplemental explains that $P \sim c \exp[-C*(1-c)]$ where as in Hibler 79, $P \sim h \exp[-C*(1-c)]$. c and h are not linearly related year-round so it is not reasonable to simply swap one for the other. Was the strength parameterization actually coded with c in place of h ? And if so, what if it were not? How would Figure S3 look compared to h rather than c ? Isn't h even more important? Is the critical concentration really a critical thickness?

Have ocean-ice heat fluxes of 30 W/m² ever been measured in the central Arctic where the sea ice concentration is ~80%? Has anyone ever found that heat fluxes are larger than average in anticyclones. Presumably the authors meant an anticyclonic ocean eddy, but it was not my first assumption. Please note whether the circulations mentioned are in the ocean, ice or atmosphere.

A paper that is relevant to this one is by David Holland in J. Climate 2001 "An Impact of Subgrid-Scale Ice-Ocean Dynamics on Sea-Ice Cover"

Specific points in the Main section

I believe the first paper to discuss the speed up in sea ice with thinning is Rampal et al (2011). They frame the relationship as a positive feedback and therefore should be reviewed and cited.

"On interannual and longer timescales, climate projection models tend to overestimate sea ice extent I don't understand what timescale has to do with a the claimed bias. I also don't think it is true that models tend to overestimate sea ice extent. There are many that underestimate it, such as the GISS model in CMIP5.

What is meant by "remain largely unconstrained on basin scales" in paragraph one? Most things are unconstrained in climate simulations since there is no data assimilation and the initial conditions are long since forgotten.

Please add "Nearly always" to "Climate models do not resolve ice-ocean interactions at eddy scales" in

paragraph two. An exception is PRIMAVERA with FESOM.

"sea distributions"? presumably the word "ice" is missing.

The statement "enhancement in the correlation between the ice and ocean vorticity in MIZs" needs to say what the enhancement is relative to.

Specific points in the Methods section

What is the resolution of the atmospheric forcing and how does it influence the simulation to have it so comparably smooth?

The authors argue that VP has been found to be appropriate for the seasonal ice zones, but then they analyze their model in the perennial zones. What are the consequences of the VP assumption at such fine resolution where the model resolution is smaller than the floe size?

Something is wrong with this the english in this sentence: "the model only marginally permitting the development of submesoscale flows...". I think permitting should be permits.

Fig 2c why are units of volume in m? And is the inset an anomaly? How can the thickness change so rapidly during a cyclone? By what mechanism?

Fig 2 caption "correspondingly" is used where "respectively" is meant. The xlabel on the panels came out strangely.

Fig 4 fix the aspect ratios of the map projections and order winter and summer the same as in Fig 3.

Reviewers' Comments:

Reviewer #1:

Remarks to the Author:

The authors have responded carefully to the comments from the first review and the manuscript is improved. My general comments from the first review have been dealt with fine. Most of the specific comments are also dealt with fine although there are still various bugs. There are a few new (minor) issues to address. I recommend a minor revision of the manuscript and then acceptance once these issues are fixed.

Main Comment:

Line 186: The new quantitative estimate of the feedback strength on page 4 is helpful. The estimate of the feedback rate is $4 \times 10^{-9} \text{ s}^{-1}$, or a timescale of 7.9 years. This is a slow positive feedback. It's unlikely to be important because it will probably be swamped by faster processes (for example as suggested at lines 202-208). This questions the importance of one of the main messages of the paper about the positive feedback process missing from climate models (e.g. Abstract, Discussion). I think these issues should be tackled directly in the Discussion. Currently, the Discussion doesn't refer to the feedback timescale. Specifically, address the question: With such a slow growth rate how might the feedback be important? For instance, talk about the sense of the feedback (accelerates sea ice loss in MIZs). Is this consistent with the sea ice biases seen in climate models MIZs (the "prediction errors" mentioned at line 14)? The 7.9 year timescale suggests that the feedback would give about 7% more sea ice loss in an MIZ in one melt season (from $\exp(0.5/7.9)$). How does that compare to uncertainty in annual ice loss from uncertainty in air/sea fluxes over the melt season, and from uncertainties in other processes supplying heat to the MIZ? I think a candid discussion of these issues will enhance the value of the study to a wider range of climate scientists.

Minor Comments Main Text

Line 93: It says "sea ice is continuously advected by atmospheric winds, such that the locally-enhanced heating by anticyclones impacts a region of sea ice that may be larger than the eddies themselves." This still doesn't make sense to me. The ice vorticity is well correlated with the upper ocean vorticity, but the ice is also being advected by the wind so that the ice velocity is different to the ocean velocity. How might that happen kinematically? Presumably, there's some tacit timescale separation. What evidence is there that this process occurs? Consider cutting this line.

Line 98, 105: "Turbulence" is used in inconsistent and ambiguous ways. Be specific with characteristic properties of the turbulence.

Line 122: Mention the significance of the k^{-1} kinetic energy spectrum.

Line 132: It says the "ice-free submesoscale flows...are characterized by a k^{-2} spectrum". How is this seen in Figure 3g? Summer shows k^{-4} not k^{-2} . Also cite to support the -2 slope.

Line 140 and several other places: Figures "4a, 5a, 5b, 5c" are called out but they don't appear in the manuscript.

Minor Comments Supplement

Line 66: Justify $u_i = u_o = 0$. These zero speeds contradict geostrophy in line 37.

Line 87 which justifies equation (15): Why should the 2 rheology terms in equations (13) and (14) be

of the same order? What evidence is there that this is true?

Line 94: It says "writing the momentum equation we arrive at...". Which momentum equation? Equation (1)? I don't see how to arrive at (16) and (17).

Line 122: Why only show $C^* = 5, 10, 20$ if the canonical value is 20? Why not show a higher value too? This comment was made in the first review. Sea ice modelers will appreciate seeing variations in the C^* parameter in both senses. If the behavior is monotonic (as stated in the rebuttal), why show both $C^*=5$ and 10?

Typos, bugs etc. Main Text

Line 21: "unconstrained" should read "unknown" or "uncertain" (the fluxes aren't physically unconstrained).

Line 39: Clarify "fine-scale oceanic features" by giving a characteristic length.

Line 53 (the line numbering is corrupted here): I think "Figure 1a" should be "Figure 1b" (and 1c?) in items (i) and (ii).

Eq(1): Where does the length scale L come from?

Line 54: Why is the Coriolis parameter " f " a vector?

Line 152: Fix "and and"

Line 264 (and several other places): "LLC" should be "LLC4320" because there are other "LLC" grid models with much coarser resolution than LLC4320.

Line 266: "capture" -> "captures"

Line 269: " ρ_o " is displayed wrong.

Figure 2d subpanels: What is the depth(s) of these results?

Typos, bugs etc. Supplement:

Line 28: Fix "resul"

Line 42: "m" in the formula should be m_i (?)

Equation (6): ρ should be ρ_o (?). Also write the equation with \approx not =

Equation (8): What is subscript k ?

Equation (9): ϵ should have subscripts I think.

Line 79, 80: The formulas should concern $\nabla \cdot \sigma$, not $\nabla \sigma$ (otherwise, explain).

Line 84: It talks about "a non-zero force due to shearing of ocean currents". You mean ice currents? The viscous term you refer to is due to ice viscosity, not water viscosity.

Line 96: Fix "h=1! m"

Reviewer #2:

None

Revision 2

“Heavy footprints of upper-ocean eddies on weakened Arctic sea ice in marginal ice zones”: response to reviewers’ comments.

G.E. Manucharyan and A. F. Thompson

Dec 2021

We thank the reviewer for their thorough reviews of our manuscript and for the valuable suggestions that have improved the content and presentation. We provide detailed point-by-point responses to all the comments below, and we have implemented all of the recommended changes in the manuscript.

Reviewer 1

The authors have responded carefully to the comments from the first review and the manuscript is improved. My general comments from the first review have been dealt with fine. Most of the specific comments are also dealt with fine although there are still various bugs. There are a few new (minor) issues to address. I recommend a minor revision of the manuscript and then acceptance once these issues are fixed.

Main Comment:

Line 186: The new quantitative estimate of the feedback strength on page 4 is helpful. The estimate of the feedback rate is $4 \times 10^{-9} s^{-1}$, or a timescale of 7.9 years. This is a slow positive feedback. It’s unlikely to be important because it will probably be swamped by faster processes (for example as suggested at lines 202-208). This questions the importance of one of the main messages of the paper about the positive feedback process missing from climate models (e.g. Abstract, Discussion). I think these issues should be tackled directly in the Discussion. Currently, the Discussion doesn’t refer to the feedback timescale. Specifically, address the question: With such a slow growth rate how might the feedback be important? For instance, talk about the sense of the feedback (accelerates sea ice loss in MIZs). Is this consistent with the sea ice biases seen in climate models MIZs (the “prediction errors” mentioned at line 14)? The 7.9 year timescale suggests that the feedback would give about 7% more sea ice loss in an MIZ in one melt season (from $\exp(0.5/7.9)$). How does that compare to uncertainty in annual ice loss from uncertainty in air/sea fluxes over the melt season, and from uncertainties in other processes supplying heat to the MIZ? I think a candid discussion of these issues will enhance the value of the study to a wider range of climate scientists.

We have now expanded the discussion of the feedback strength, found in the last paragraph of the Section “Estimating the eddy heat flux feedback.” To summarize briefly here, we acknowledge in the new text that that eddy feedback timescale operates on time scales longer than the seasonal cycle. Furthermore, the impact on the ocean-ice heat flux, $O(1-2 \text{ W m}^{-2})$ is small compared to radiative fluxes from the atmosphere. Nevertheless, the eddy flux feedback will have its largest amplitude in regions with large changes in sea ice concentration, *e.g.* marginal ice zones, and thus will amplify the seasonal sea-ice cycle. These changes will have additional feedbacks related to coupling with the atmosphere that are not captured in our relatively simple scaling. Furthermore, these small changes in ocean-ice heat flux, if persistent, can impact the equilibrium sea ice thickness. Finally, following the analysis you suggest for estimating sea ice loss across a single season, we would argue that a $O(10\%)$ change is non-negligible, especially if it leads to an expansion of the marginal ice zone and a growth in the area over which the eddy feedback makes a significant contribution

to sea ice dynamics.

Minor Comments Main Text:

Line 93: It says “sea ice is continuously advected by atmospheric winds, such that the locally-enhanced heating by anticyclones impacts a region of sea ice that may be larger than the eddies themselves.” This still doesn’t make sense to me. The ice vorticity is well correlated with the upper ocean vorticity, but the ice is also being advected by the wind so that the ice velocity is different to the ocean velocity. How might that happen kinematically? Presumably, there’s some tacit timescale separation. What evidence is there that this process occurs? Consider cutting this line.

Assuming the translational velocity is uniform over a broad spatial scale, and thus does not make a strong contribution to the local vorticity, the sea ice can be strongly correlated with the underlying ocean velocity, even if the ice is moving between individual coherent eddies. Thus, if the atmosphere-ice stress does not contain significant vorticity then the sea ice vorticity is only affected by the ocean vorticity. We believe our description in Line 93 is accurate. We have however removed this sentence as it wasn’t overly necessary to the discussion in that paragraph.

Line 98, 105: “Turbulence” is used in inconsistent and ambiguous ways. Be specific with characteristic properties of the turbulence.

We have fixed our usage of the word turbulence, in particular we have substituted “mesoscale and submesoscale turbulence” where it was previously ambiguous.

Line 122: Mention the significance of the k^{-1} kinetic energy spectrum.

We now mention that the k^{-1} slope of the spectrum indicates that the kinetic energy field has a significant amount of its power at smaller scales, especially compared with the mesoscale turbulence with k^{-3} spectrum. The discussion of the spectral slopes is presented in the next paragraph.

Line 132: It says the “ice-free submesoscale flows... are characterized by a k^{-2} spectrum”. How is this seen in Figure 3g? Summer shows k^{-4} not k^{-2} . Also cite to support the -2 slope.

We clarified that the k^{-2} submesoscale spectrum is commonly observed in temperate oceans (not in the Arctic Ocean) and added citations.

Line 140 and several other places: Figures “4a, 5a, 5b, 5c” are called out but they don’t appear in the manuscript.

Those figures appear at the end of the paper; in total there are 5 figures now.

Minor Comments Supplement

Line 66: Justify $u_i = u_o = 0$. These zero speeds contradict geostrophy in line 37.

We now address this apparent inconsistency. We assume that the along-frontal direction is y , so the cross-frontal velocity U is substantially smaller than the along-front velocity V . In Line 37, we now specify that the geostrophic velocity is $fV_o = \phi_x$ instead of $fU_o = -\phi_y$.

Line 87 which justifies equation (15): Why should the 2 rheology terms in equations (13) and (14) be of the same order? What evidence is there that this is true?

The two terms represent the pressure part and the viscous part of the sea ice stress tensor, which, following the viscous-plastic rheology (Hibler 1979), are of similar magnitude since both scale with the ice strength such that the yield curve is an ellipse with a relatively low eccentricity of 2. In other words, the normal and shear stresses are of the same order of magnitude when the ice is yielding. The evidence for this was shown in [1].

Line 94: It says “writing the momentum equation we arrive at...”. Which momentum equation? Equation (1)? I don’t see how to arrive at (16) and (17).

We now provide additional explanation for how the general ice momentum Equation (1) can be simplified for the case of free-drifting sea ice over the ocean without the atmospheric forcing. In brief, if ice is given an initial velocity in a stationary ocean, then in the absence of external winds, the ice velocity will approach

the ocean velocity over a characteristic timescale of about an hour.

Line 122: Why only show $C^* = 5, 10, 20$ if the canonical value is 20? Why not show a higher value too? This comment was made in the first review. Sea ice modelers will appreciate seeing variations in the C^* parameter in both senses. If the behavior is monotonic (as stated in the rebuttal), why show both $C^*=5$ and 10?

The three shown cases of $C^*=\{20,10,5\}$ together with the functional form used to define the ice strength in the VP rheology, $P^* \sim \exp(-C^*(1-c))$, demonstrate the monotonicity of $c_{cr} = c_{cr}(C^*)$. The canonical value of $C^* = 20$ provides a relatively narrow range of concentrations for which the ice and ocean vorticity are not correlated ($0.8 < c < 1$). For higher C values the concentration range will become even narrower and it will be hard to see on the figure; for example, for $C^* = 30$, the concentration must exceed 0.95 to have sea ice not responding to ocean eddies. We thus keep the figures as currently formatted for presentation purposes as it is sufficient to convey the main point of our paper that the critical sea ice concentration depends on the details of sea ice rheology, specifically on the parameter C^* of the VP rheology.

Typos, bugs etc. Main Text

Line 21: “unconstrained” should read “unknown” or “uncertain” (the fluxes aren’t physically unconstrained).

Changed to uncertain.

Line 39: Clarify “fine-scale oceanic features” by giving a characteristic length.
We added $O(10 \text{ km})$ here.

Line 53 (the line numbering is corrupted here): I think “Figure 1a” should be “Figure 1b” (and 1c?) in items (i) and (ii).

In the previous revision we changed the color scheme of Figure 1a and this caused inconsistency. We have adjusted the description according to the new figure 1.

Eq(1): Where does the length scale L come from?

It was introduced to express the scaling law in terms of vorticity $\zeta \sim U/L$ instead of velocity. We added this clarification to the text.

Line 54: Why is the Coriolis parameter “ f ” a vector?

We now express \mathbf{f} as $f\mathbf{k}$ where \mathbf{k} is unit vector pointing up.

Line 152: Fix “and and”

Fixed.

Line 264 (and several other places): “LLC” should be “LLC4320” because there are other “LLC” grid models with much coarser resolution than LLC4320.

Fixed.

Line 266: “capture” -> “captures”

Fixed.

Line 269: “ ρ_o ” is displayed wrong.

Fixed.

Figure 2d subpanels: What is the depth(s) of these results?

We added that the vorticity is evaluated at a depth of 20 meters.

Typos, bugs etc. Supplement:

Line 28: Fix “result”

Fixed.

Line 42: “m” in the formula should be m_i (?)

Fixed.

Equation (6): ρ should be ρ_o (?). Also write the equation with \approx not =

We have fixed the density issue. We have chosen to keep the equal sign in equation (6) based on the stated the assumptions of steady state and unidirectional flow.

Equation (8): What is subscript k ?

We added the description of what those indices mean; in this case, k is simply used as a dummy index to denote a summation of diagonal elements of a matrix.

Equation (9): ϵ should have subscripts I think.

Fixed.

Line 79, 80: The formulas should concern $\nabla \cdot \sigma$, not $\nabla \sigma$ (otherwise, explain).

Indeed, thank you for catching this – it is fixed.

Line 84: It talks about “a non-zero force due to shearing of ocean currents”. You mean ice currents?

The viscous term you refer to is due to ice viscosity, not water viscosity.

Fixed.

Line 96: Fix “h=1! m”

Fixed.

Thank you for spotting all the typographic issues! We have fixed all of them.

References

- [1] Georgy E Manucharyan and Andrew F Thompson. Submesoscale sea ice-ocean interactions in marginal ice zones. *Journal of Geophysical Research: Oceans*, 122(12):9455–9475, 2017.

Reviewers' Comments:

Reviewer #1:

Remarks to the Author:

The authors have responded thoughtfully and carefully to the review. I recommend publication of this version.